# p38-mediated phosphorylation at T367 induces EZH2 cytoplasmic localization to promote breast cancer metastasis

Talha Anwar [1,2,3,4], Caroline Arellano-Garcia[5], James Ropa[1,2], Yu-Chih Chen[3,6], Hong Sun Kim[1,3], Euisik Yoon[6,7], Sierrah Grigsby[1,2], Venkatesha Basrur [1], Alexey I. Nesvizhskii [1], Andrew Muntean[1], Maria E. Gonzalez[1,3], Kelley M. Kidwell[8], Zaneta Nikolovska-Coleska[1] & Celina G. Kleer[1,3]

Overexpression of EZH2 in estrogen receptor negative (ER-) breast cancer promotes metastasis. EZH2 has been mainly studied as the catalytic component of the Polycomb Repressive Complex 2 (PRC2) that mediates gene repression by trimethylating histone H3 at lysine 27 (H3K27me3). However, how EZH2 drives metastasis despite the low H3K27me3 levels observed in ER- breast cancer is unknown. Here we show that in human invasive carcinomas and distant metastases, cytoplasmic EZH2 phosphorylated at T367 is significantly associated with ER- disease and low H3K27me3 levels. p38-mediated EZH2 phosphorylation at T367 promotes EZH2 cytoplasmic localization and potentiates EZH2 binding to vinculin and other cytoskeletal regulators of cell migration and invasion. Ectopic expression of a phospho-deficient T367A-EZH2 mutant is sufficient to inhibit EZH2 cytoplasmic expression, disrupt binding to cytoskeletal regulators, and reduce EZH2-mediated adhesion, migration, invasion, and development of spontaneous metastasis. These results point to a PRC2-independent non-canonical mechanism of EZH2 pro-metastatic function.

[1] Department of Pathology, University of Michigan Medical School, Ann Arbor, MI 48109, USA. [2] Molecular Cellular and Pathology Training Program, University of Michigan, Ann Arbor, MI 48109, USA. [3] Rogel Cancer Center, University of Michigan, Ann Arbor, MI 48109, USA. [4] Medical Scientist Training Program, University of Michigan, Ann Arbor, MI 48109, USA. [5] Michigan Post-baccalaureate Research Education Program, University of Michigan, Ann Arbor, MI 48109, USA. [6] Department of Electrical Engineering and Computer Science, University of Michigan, Ann Arbor, MI 48109, USA. [7] Department of Biomedical Engineering, University of Michigan, Ann Arbor, MI 48109, USA. [8] Department of Biostatistics, University of Michigan, Ann Arbor, MI 48109, USA. Correspondence and requests for materials should be addressed to C.G.K. (email: kleer@umich.edu)

The overwhelming majority of breast cancer deaths occur due to metastasis. Breast cancer patients with distant metastases at the time of diagnosis have significantly worse prognosis with a 5-year survival rate of 23.4%[1]. New effective strategies for inhibiting metastatic spread or blocking the growth of established distant metastasis are needed.

Tumor cells must undergo fundamental changes to their identity to acquire the traits needed for dissemination to distant sites. Dysregulation of factors governing cell type identity is a common feature of metastatic cancer. Enhancer of zeste homolog 2 (EZH2) has been shown to regulate these processes through epigenetic silencing. Our lab and others have demonstrated that EZH2 is overexpressed in human solid and hematopoietic malignancies[2–5]. In breast cancer, EZH2 overexpression is significantly associated with the estrogen receptor-negative (ER-) subtype and worse clinical outcome[2]. As the catalytic subunit of the Polycomb repressive complex 2 (PRC2), the methyltransferase EZH2 deposits trimethyl marks on histone tails of lysine 27 of histone H3 (H3K27me3) to effect transcriptional repression. However, the high levels of EZH2 observed in ER- tumors are associated with low H3K27me3[6–8], suggesting that the oncogenic function of EZH2 may rely on mechanisms other than repression of tumor suppressor genes, which are currently unknown.

Metastatic progression also involves tight regulation of the cellular responses elicited by the microenvironment. p38 MAPK proteins are critical in signaling cascades that transduce extracellular stimuli—inflammation, hypoxia, growth factors, and cytokine stimulation—into biological responses through proline-directed serine/threonine phosphorylation of target substrates commonly involved with gene expression regulation. The most abundant p38 family member, p38α (also known as MAPK14), has a well-documented, albeit complex role in cancer, exerting cell-type dependent tumor-suppressive or tumor-promoting functions[9]. In the breast, p38α promotes breast cancer progression[10–12], and high levels of active p38 MAPK are biomarkers of poor prognosis[9,13,14]. However, how p38α MAPK activity induces breast cancer progression remains ill-defined.

We have demonstrated that EZH2 and p38α interact in aggressive ER- breast cancer cells[15], and EZH2 has been shown to undergo p38α-mediated T367 phosphorylation during muscle regeneration[16]. However, direct demonstration that p38α phosphorylates EZH2 in solid tumors, the biological consequences of EZH2 T367 phosphorylation in breast cancer, and the mechanisms of pEZH2(T367) function are still unclear. Despite evidence of cytoplasmic EZH2 in aggressive breast cancers[17], studies have focused on the nuclear functions of EZH2, and the functions of EZH2 in the cytoplasm have remained elusive.

Here, we report that EZH2 is regulated by p38-mediated T367 phosphorylation during breast cancer progression. We show that this phosphorylation event controls EZH2 subcellular localization and is sufficient to activate the metastasis promoting function of EZH2 in breast cancer. Our data reveal that pEZH2(T367) is upregulated in the cytoplasm of cancer cells in clinical samples of invasive breast carcinoma and distant metastasis in contrast with normal breast epithelium. We provide the foundation to block p38-mediated EZH2 T367 phosphorylation as a potential therapeutic strategy for metastatic breast cancer.

## Results

### pEZH2(T367) is in the cytoplasm of invasive breast cancer.
To investigate whether p38α phosphorylates EZH2 at T367 in breast cancer and study the biological relevance, we developed and validated a rabbit polyclonal anti-phosphorylated T367 EZH2 antibody. In dot blot analyses, the anti-pEZH2(T367) antibody

specifically recognized a peptide corresponding to the phosphorylated T367 site but not the unmodified peptide (Supplementary Fig. 1A). Demonstrating its specificity for the T367 phosphorylated form of the EZH2 protein, the anti-pEZH2(T367) antibody failed to detect dephosphorylated recombinant EZH2 and dephosphorylated EZH2 from breast cancer cell lysate (Supplementary Fig. 2B and C). Incubation of the antibody with the phosphorylated peptide outcompeted antibody binding in Western blot analysis of whole cell extracts of MDA-MB-231 cells and in immunohistochemistry of breast cancer tissue samples, further demonstrating specificity of the antibody for this site (Supplementary Fig. 1D and E). Finally, the antibody failed to recognize an unphosphorylatable threonine to alanine (T367A) mutant (Supplementary Fig. 1F), suggesting that the antibody is specific for phosphorylation at the T367A site.

We evaluated pEZH2(T367) protein expression in situ by immunohistochemistry in a wide range of breast tissue samples from 146 patients, including normal breast ($n = 19$), invasive carcinomas ($n = 104$), and distant metastasis ($n = 23$) arrayed in high density tissue microrrays in triplicate (Supplementary Table 1). Although in normal lobules pEZH2(T367), if present, was localized to the nucleus, pEZH2(T367) was expressed in the cytoplasm of invasive breast cancer cells (Fig. 1a). The frequency of cytoplasmic pEZH2(T367) increased significantly with breast cancer progression, as it was absent in normal lobules and detected in 57% of invasive carcinomas and in 74% of breast cancer distant metastasis (Chi-square $p = 0.0001$, Fig. 1b). In the 104 primary invasive carcinomas, high cytoplasmic pEZH2(T367) was significantly associated with higher histological grade ($p = 0.028$), ER- ($p = 0.0003$), PR- ($p = 0.0002$), and triple negative status ($p = 0.0006$) (Supplementary Table 2).

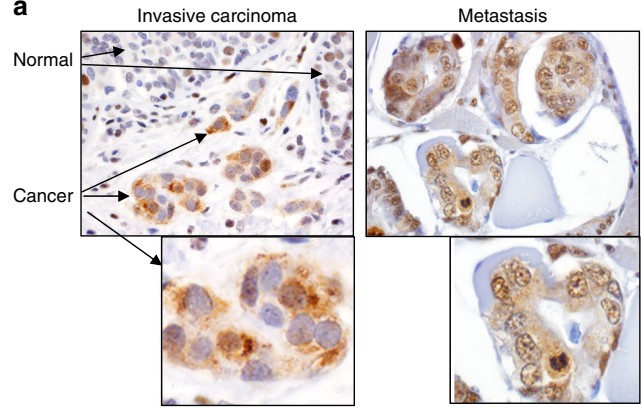

**b**

| Diagnosis | Cyto pEZH2 | |
|---|---|---|
| | Low, $n$ (%) | High, $n$ (%) |
| Normal | 19 (100%) | 0 (0%) |
| Invasive ca | 45 (43%) | 59 (57%) |
| Metastasis | 6 (26%) | 17 (74%) |

Chi-square $p < 0.00001$

**Fig. 1** Phosphorylated EZH2 (T367) is expressed in the cytoplasm of invasive breast carcinoma and distant metastases. **a** Immunohistochemical analysis of pEZH2(T367) expression using a specific antibody in human tissue samples of 193 patients. Pictures show a representative invasive breast carcinoma with adjacent normal breast (left) and metastasis (right) (×400 magnification). Insets show expression of pEZH2(T367) in cancer cells (×600 magnification). **b** Results are tabulated. Cytoplasmic pEZH2 is significantly associated with invasive carcinoma and metastasis compared to normal and fibrocystic changes (Chi-square $p < 0.00001$)

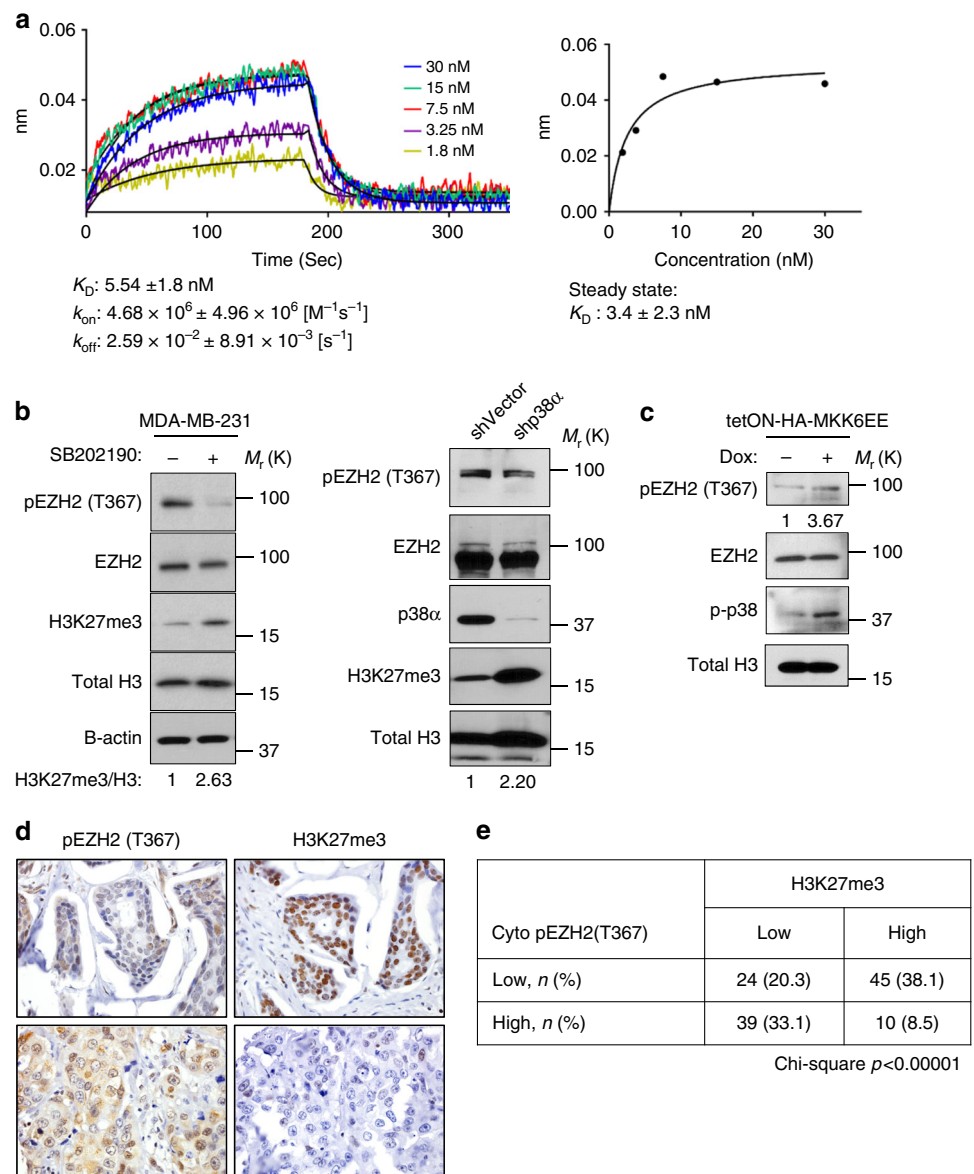

**Fig. 2** p38 phosphorylates EZH2 in breast cancer. **a** Quantitative analysis of the direct interaction between recombinant EZH2 and p38α proteins using BioLayer Interferometry. EZH2 was immobilized on a sensor chip and a concentration series of p38α protein was added. Sensorgrams and corresponding fitting curves for kinetics constants and affinity determination (left) and corresponding plot of steady state response against concentration (right) for determination of binding affinity. **b** Western blot of MDA-MB-231 cells treated with 20 μM SB202190 for 48 hr (left panel) or with p38 shRNA (right panel) to pharmacologically and genetically inhibit p38 activity, respectively. **c** Western blot of MDA-MB-231 cells transduced with dox-inducible MKK6EE to activate p38α. **d**, **e** Immunohistochemical staining of 118 samples of human invasive breast carcinomas using anti-pEZH2(T367) and anti-H3K27me3 antibodies demonstrating a significant inverse association between these protein modifications (Chi-square, $p < 0.00001$) (×400 magnification)

As is the case for clinical samples of breast cancer, in a panel of breast cell lines, we observed that aggressive and metastatic ER- breast cancer cells exhibit higher pEZH2(T367), EZH2, and p-p38 proteins compared to benign and less aggressive ER-positive breast cancer cells and nontumorigenic breast cells (Supplementary Fig. 1G).

A survey of pEZH2(T367) expression in a cohort of human normal and cancer tissues showed that cytoplasmic pEZH2(T367) is highly expressed in epithelial malignancies, including kidney and colon cancer, hepatocellular carcinoma, and thyroid carcinoma compared to normal tissues and to non-epithelial tumors (Supplementary Fig. 2).

Altogether, these results demonstrate that cytoplasmic pEZH2(T367) is expressed in invasive and metastatic breast carcinomas where it is associated with higher histological grade, a measure of tumor aggressiveness, ER- status, and breast cancer progression. We observe that pEZH2(T367) is upregulated in several types of human carcinoma compared to corresponding normal tissue.

**p38 phosphorylates EZH2 at T367 in breast cancer**. To examine the function and regulation of pEZH2(T367) in breast tumorigenesis, we used MDA-MB-231, MDA-MB-468, and SUM159 breast cancer cell lines which exhibit p38-EZH2 binding, high endogenous levels of p-p38[15], and high pEZH2(T367) (Supplementary Fig. 1G). Quantitative Bio-Layer Interferometry (BLI) data with recombinant proteins indicated a strong binding affinity of EZH2 for p38α with binding affinity KD of 5.54 nM and

kinetic constants: association rate of $k_{on} = 4.68 \times 10^6$ $(M^{-1} s^{-1})$, and dissociation rate, $k_{off} = 2.59 \times 10^{-2}$ $[s^{-1}]$. Affinity constant obtained from the kinetic analysis was in excellent agreement with the steady state analysis, KD of 3.4 nM, confirming the direct and strong binding interaction between EZH2 for p38α (Fig. 2a).

We next investigated the effect of p38α-mediated T367 phosphorylation on EZH2 function, through complementary and independent approaches to inhibit and to activate p38 MAPK. Both p38α knockdown using stable lentiviral-mediated short hairpin RNA interference (shRNA) and chemical inhibition of p38α/β activity with SB202190 significantly reduced pEZH2 (T367) protein in breast cancer cells without affecting total levels of EZH2 and was accompanied by increased H3K27me3 levels (Fig. 2b). Activation of p38α, the most abundant isoform, occurs through a dual phosphorylation event at T180/T182 by upstream kinases MKK3 and MKK6[18]. Transduction with an inducible, constitutively activated mutant MKK6 (MKK6EE)[19] resulted in increased p-p38 and pEZH2(T367) (Fig. 2c).

Demonstrating the significance of these results, clinical samples of invasive breast carcinoma showed a significant inverse association between cytoplasmic pEZH2(T367) and H3K27me3 levels (Chi-square $p < 0.00001$, Fig. 2d, e). Taken together, these data show a direct interaction between p38α and EZH2, that p38α phosphorylates EZH2 at T367 in breast cancer cells and human tissues, and suggest that p38α-mediated phosphorylation reduces EZH2-mediated trimethylation of H3K27.

**p38 T367 phosphorylation of EZH2 promotes cytoplasmic localization.** Dox-induced MKK6 activation of p38α was sufficient to promote cytoplasmic localization of GFP-EZH2 expressed in MDA-MB-231 cells compared to controls (Fig. 3a). To directly investigate the relevance of phosphorylation at T367 to the subcellular localization of EZH2, we generated a GFP-tagged EZH2 mutant by replacing T367 with Ala (T367A). In MDA-MB-231, SUM159, and MDA-MB-468 cells, ectopic GFP-EZH2-T367A was nuclear and showed reduced localization to the cytoplasm (Fig. 3b, Supplementary Fig 3A), demonstrating that T367 phosphorylation is required for the cytoplasmic expression of EZH2. Further supporting this observation, fractionation of MDA-MB-231 and MDA-MB-468 cells showed an enrichment of pEZH2(T367) in the cytoplasmic compartment and an absence in the chromatin-bound compartment compared to total EZH2 (Supplementary Fig 3B).

To test the relevance of cytoplasmic EZH2 to the neoplastic functions of breast cancer we developed an EZH2 mutant lacking the nuclear localization domain (ΔNLS-EZH2) (Fig. 3c). To avoid the contribution of endogenous EZH2 we first generated MDA-MB-231 cells with stable 3′UTR EZH2 knockdown followed by rescue with full length (WT-EZH2) and ΔNLS-EZH2 mutant adenoviral constructs (Fig. 3d). ΔNLS-EZH2 expression was cytoplasmic and retained the ability to interact with SUZ12 and EED but had no effect on H3K27me3 (Supplementary Fig. 3C–D). Rescue ΔNLS-EZH2 was sufficient to restore the reduced invasion and migration of MDA-MB-231 shEZH2 breast cancer cells to similar levels that WT-EZH2 (Fig. 3e, f). Collectively, these data show that T367 phosphorylation is required for EZH2 cytoplasmic localization in breast cancer cells, and reveal that cytoplasmic EZH2 expression is sufficient to promote breast cancer cell migration and invasion.

**pEZH2(T367) is essential for invasion and metastasis.** We hypothesized that in the cytoplasm, pEZH2(T367) may regulate the migratory and invasive abilities of breast cancer cells, and set out to rescue the expression of WT-EZH2 and the

phospho-deficient T367A-EZH2 mutant in MDA-MB-231, -468, and SUM159 cells with EZH2 3′UTR knockdown (Fig. 4a, Supplementary Fig. 4A). In contrast with WT-EZH2, ectopic expression of T367A-EZH2 was unable to rescue breast cancer cell migration and invasion (Fig. 4b–d, Supplementary Fig. 4B–C). Based on the observation that T367A-EZH2 expressing cells adhered strongly to the substrate during culture, we reasoned that EZH2 phosphorylation may regulate this function. Our studies demonstrate that T367A-EZH2 significantly increased attachment of breast cancer cells in adhesion assays compared to WT-EZH2 (Fig. 4e, Supplementary Fig. 4D).

Ectopic expression of T367A-EZH2 was able to restore the proliferative abilities of knockdown-rescue cells as effectively as WT-EZH2 (Fig. 4f, Supplementary Fig. 4E–F), demonstrating that T367 phosphorylation is dispensable for EZH2 proliferative functions. Our data also reveal that T367 phosphorylation did not significantly affect the stability of EZH2 protein or its ability to bind with other PRC2 proteins SUZ12 and EED (Supplementary Fig. 5A–B).

As expected based on our previous results[20], MDA-MB-231 expressing shRNA against EZH2 formed smaller tumors than controls. Validating the dispensable role for T367 phosphorylation on cell proliferation observed in vitro, primary tumors formed by T367-EZH2 and WT-EZH2 were of similar size (Fig. 5a). Despite no differences in primary tumor growth kinetics (Fig. 5b), T367A-EZH2 signficantly reduced the lung metastatic burden and ability to metastasize compared to WT-EZH2 (Fig. 5c–e). Altogether, these data document that T367 phosphorylation is critical for the metastasis-promoting function of EZH2 in breast cancer.

**Phosphorylation promotes binding to cytoplasmic proteins.** Based on the significant association between cytoplasmic pEZH2 (T367) and breast cancer invasion and metastasis in clinical samples, and its critical role in promoting breast cancer progression, we hypothesized that pEZH2(T367) may interact with cytoplasmic and cytoskeletal regulatory proteins. To map pEZH2 (T367) interactors, whole cell lysates of knockdown-rescue MDA-MB-231 cells expressing FLAG vector, FLAG-WT, or FLAG-T367A were affinity purified using FLAG immunoprecipitation. We then performed liquid chromatography/tandem mass spectrometry (LC/MS/MS) analysis of proteins that coprecipitated with wild-type or mutant EZH2 from MDA-MB-231 cell lysates. We scored wild-type and mutant EZH2 interactions using MS/ MS spectral counting to calculate the Significance Analysis of Interactome (SAINT) probability and empirical fold-change scores (FC) for each prey protein using the CRAPome resource[21]. Validating the robustness of the assay, we calculated the SAINT probabilities of the three biological replicates for known interactors of EZH2 (Supplementary Table 3). Comparative analyses revealed 45 proteins that coprecipitated significantly more with FLAG-EZH2 than with FLAG-T367A, suggesting a requirement for T367 phosphorylation in regulating these potential interactions (Fig. 6a, b). Consistent with our subcellular localization studies, DAVID functional analysis showed enrichment for FLAG–EZH2 interactors in the cytoplasm and actin-binding functional annotations compared to FLAG-T367A (Fig. 6c), which included important regulators of cell migration, adhesion, and invasion (Fig. 6d).

**pEZH2(T367) binds with vinculin in ER- breast cancer cells.** Among the top differential interactors of pEZH2(T367) in the actin-binding set was vinculin, a cytoplasmic membrane and cytoskeletal protein found at focal adhesions with roles in breast cancer migration and invasion[22,23]. We validated and investigated

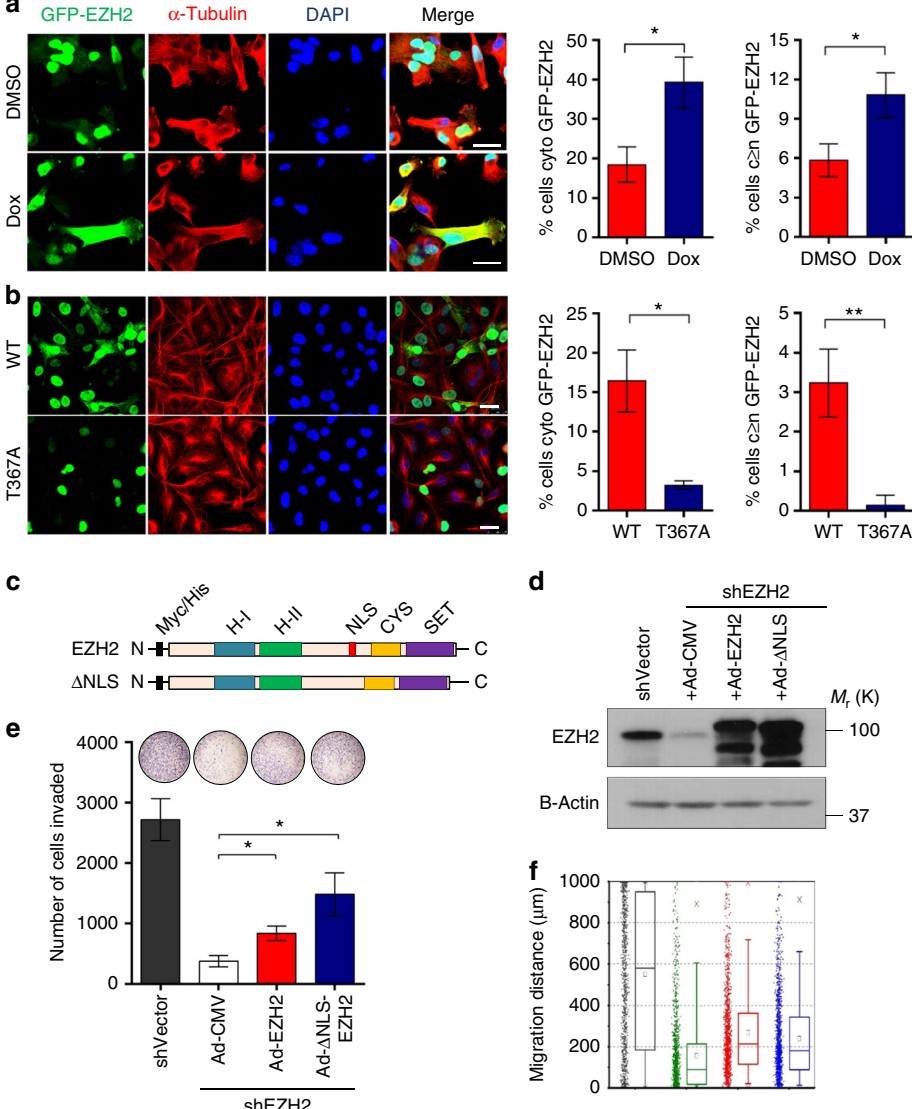

**Fig. 3** p38-mediated phosphorylation at T367 promotes EZH2 cytoplasmic localization and is sufficient for migration and invasion of breast cancer cells. **a** Immunofluorescence images of MDA-MB-231 cells transduced with lentiviruses to express GFP-EZH2 and a dox-inducible, constitutively active MKK6 kinase (MKK6EE). The percentage of non-mitotic cells expressing cytoplasmic EZH2 and cytoplasmic GFP-EZH2≥nuclear expression was quantified for >50 cells from three fields. Scale bars: 25 μm. **b** Immunofluorescence images of MDA-MB-231 cells transduced with lentivirus to express GFP-EZH2 wild-type or T367A protein. The percentage of non-mitotic cells expressing cytoplasmic EZH2 was quantified for >50 cells from three fields. Statistical analyses were performed using student's t-test. Scale bars: 25 μm. Data for **a**, **b** shown as mean±SD and are representative from an independent experiment that was repeated with three biological replicates, each with at least three technical replicates. Statistical analyses were performed using student's t-test. **c** Schematic diagram of myc-tagged EZH2 and nuclear localization signal (NLS) mutant (top left). **d** Western blot analysis of MDA-MB-231 cells showing EZH2 knockdown after lentiviral transduction with control shRNA (shVector) or 3′ UTR EZH2-targeting shRNA (shEZH2) and rescue with myc-tagged Ad-EZH2 or Ad-ΔNLS mutant (top right). Ad-CMV, adenovirus control vector. **e** Cell invasion assay of cells in **d** using a reconstituted Boyden basement membrane invasion chamber assay. Data are from at least three independent experiments carried out in at least triplicate and are presented as mean±SD. **f** Cell migration assays were performed in cells described in **d** using a high-throughput microfluidic migration platform to measure migration distance after 24 h. Data were collected from four replicates (a total of 1200 channels) were performed. Box graphs were plotted using Origin 9.0. The bottom and top of the box are the first and third quartiles, and the band inside the box is always the second quartile (the median). The ends of the whiskers represent the 5th percentile and the 95th percentile. The square inside the box indicates the mean, and the x outside the box indicates the minimum and maximum of all of the data. *$p \leq 0.05$; **$p \leq 0.01$

the mechanistic details of the interaction between pEZH2(T367) and vinculin using multiple independent and complementary strategies. By immunofluorescence, pEZH2(T367) colocalized with vinculin in the cytoplasm of MDA-MB-231, 468, and SUM159 cells (Supplementary Fig. 6A). These data are further supported by immunofluorescence studies using GFP-T367A-EZH2 and ΔNLS-EZH2 in MDA-MB-231 cells (Supplementary

Figs. 6B–C). Using proximity ligation assays (PLA) with confocal imaging, we detected pEZH2(T367)-vinculin interaction (<40 nm apart) in the cytoplasm of breast cancer cells (Fig. 7a, Supplementary Fig. 6D).

To investigate the details of the EZH2-vinculin interaction, we evaluated real-time interactions and quantified the binding affinity using BLI. Recombinant vinculin protein showed strong

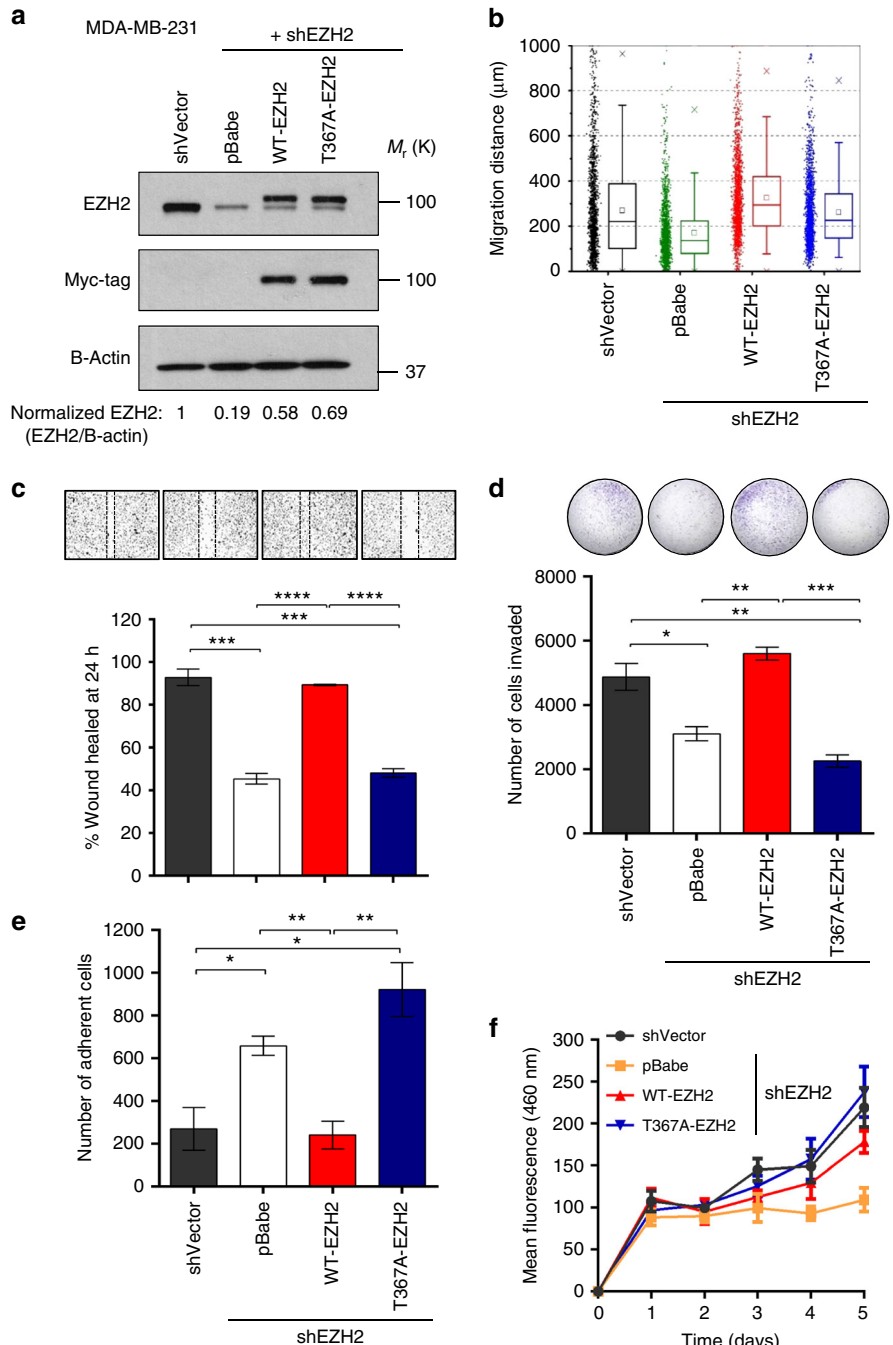

**Fig. 4** pEZH2(T367) is promotes breast cancer cell migration, invasion, and adhesion without affecting cell proliferation. **a** Western blot analysis of MDA-MB-231 breast cancer cells showing EZH2 knockdown after lentiviral transduction with control shRNA (shVector) or 3′ UTR EZH2-targeting shRNA (shEZH2) and rescue with myc-tagged WT-EZH2 or T367A-EZH2. **b** Cells described in **a** employed in a high-throughput microfluidic migration platform. Data colleted from four independent biological replicates. Box graphs were plotted using Origin 9.0. The bottom and top of the box are the first and third quartiles, and the band inside the box is always the second quartile (the median). The ends of the whiskers represent the 5th percentile and the 95th percentile. The square inside the box indicates the mean, and the x outside the box indicates the minimum and maximum of all of the data. **c** Cells described in **a** were seeded in 12-well plates, grown to confluence, and subjected to wound healing assays. Representative images of the wound after 24 h shown above bars. **d** Reconstituted Boyden basement membrane invasive chamber assay of cells in **a**. Representative chambers after crystal violet staining shown above bars. **e** Cells described in **a** employed in a cell attachment assay. **f** Cells described in **a** subjected to time course proliferation assay using Hoescht 33258 to quantify dsDNA. Data for **c**–**f** are from at least three independent experiments carried out in at least triplicate and are presented as mean ±SD. *$p \leq 0.05$; **$p \leq 0.01$; ***$p \leq 0.005$; ****$p \leq 0.0001$

binding affinity of immobilized EZH2 with $K_D$ of 15 nM and kinetic constants: association rate of $k_{on} = 2.94 \times 10^5 + 3.80 \times 10^5$ ($M^{-1} s^{-1}$) and dissociation rate, $k_{off} = 3.85 \times 10^{-3} + 8.39 \times 10^{-4}$ ($s^{-1}$). Affinity constant obtained from the kinetic analysis was in agreement with the steady state analysis with $K_D$ value of 42 nM,

confirming the direct and strong binding interaction between EZH2 and vinculin (Fig. 7b).

To determine whether p38 activation was required for EZH2-vinculin binding in breast cancer cells, we induced p38 MAPK signaling in MDA-MB-231 cells using the tetON-HA-MKK6EE

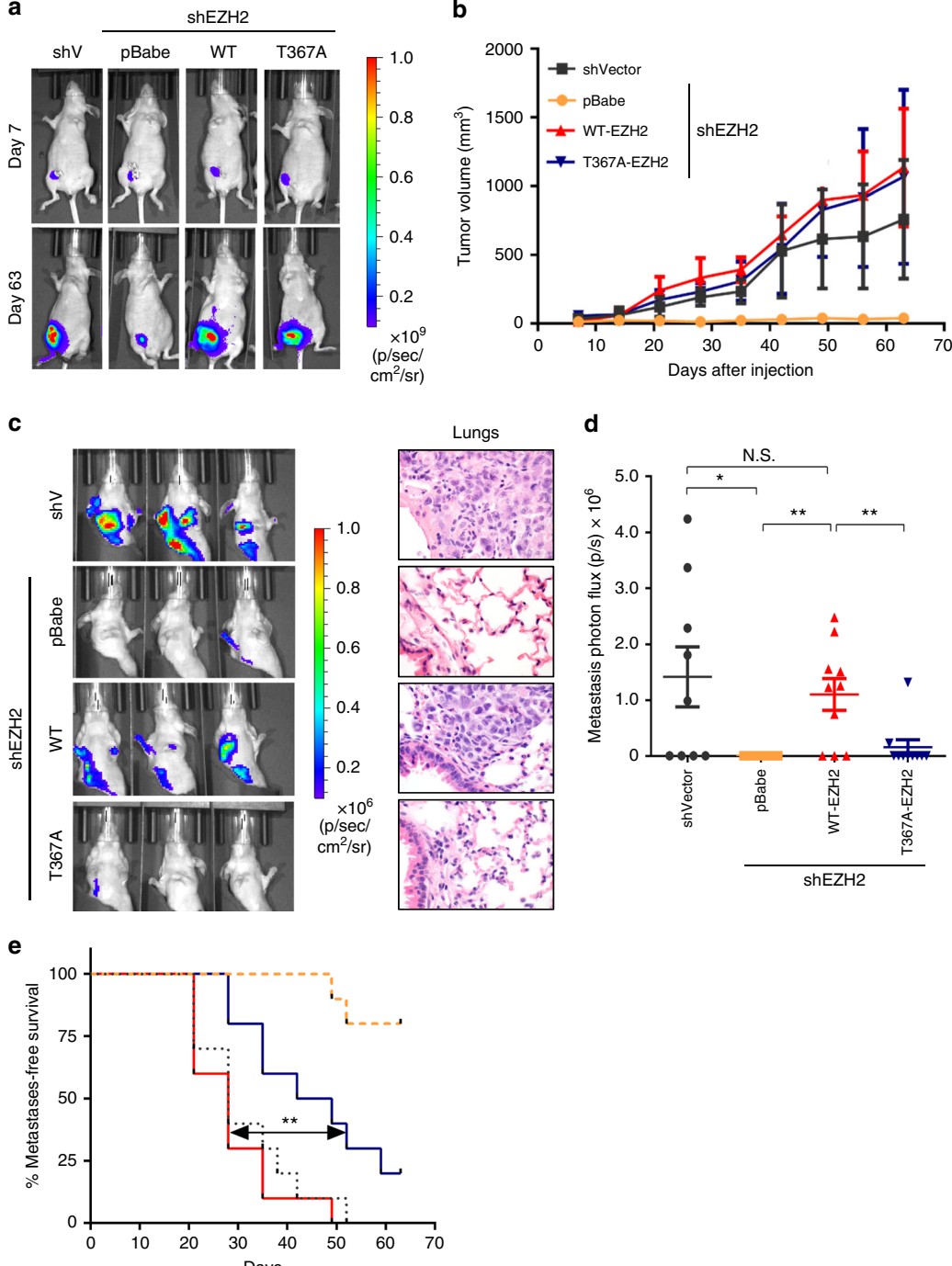

**Fig. 5** Inhibition of EZH2 T367 phosphorylation reduces breast cancer metastasis. **a** Representative bioluminescence images of primary tumors at one and nine weeks post tumor implantation. **b** Primary tumor growth curves of NOD/SCID mice orthotopically-implanted with MDA-MB-231 EZH2 knockdown rescue cells ($n = 10$ per condition) expressing GFP-Luciferase. Primary tumor growth as determined by caliper measurements, shown as mean ± SD. **c** Representative bioluminescence images of metastases (primary tumor shielded), imaged at four weeks post tumor implantation (left) with representative H&E staining of lung tissue from each of the four groups at nine weeks post implantation (right, ×600 magnification). **d** Metastatic lung burden assessed by measuring photon flux measured four weeks post tumor implantation using Live Image Pro after shielding primary tumors. Data are presented as means ± SEM. **e** Kaplan–Meier metastasis-free survival curve of mice as determined by presence of lung metastases with bioluminescence imaging showing difference between WT-EZH2 (red) and T367A-EZH2 (blue) knockdown-rescue groups. *$p \leq 0.05$; **$p \leq 0.01$

system. Activation of p38 led to an approximate 10-fold increase pEZH2(T367)-vinculin binding compared to uninduced controls (Fig. 7c).

We next investigated the consequences of the interaction between EZH2 and vinculin. Although our data demonstrate that pEZH2(T367) binds to PRC2 by co-immunoprecipitation and PLA studies, we found that vinculin does not interact with EED and SUZ12 (Supplementary Fig. 6E). Further supporting a PRC2-independent mechanism, we were unable to detect vinculin methylation after incubation with PRC2 by mass spectrometry. These data coupled with the reported role of phosphorylation in vinculin activation at sites of focal adhesions[24,25], suggested the

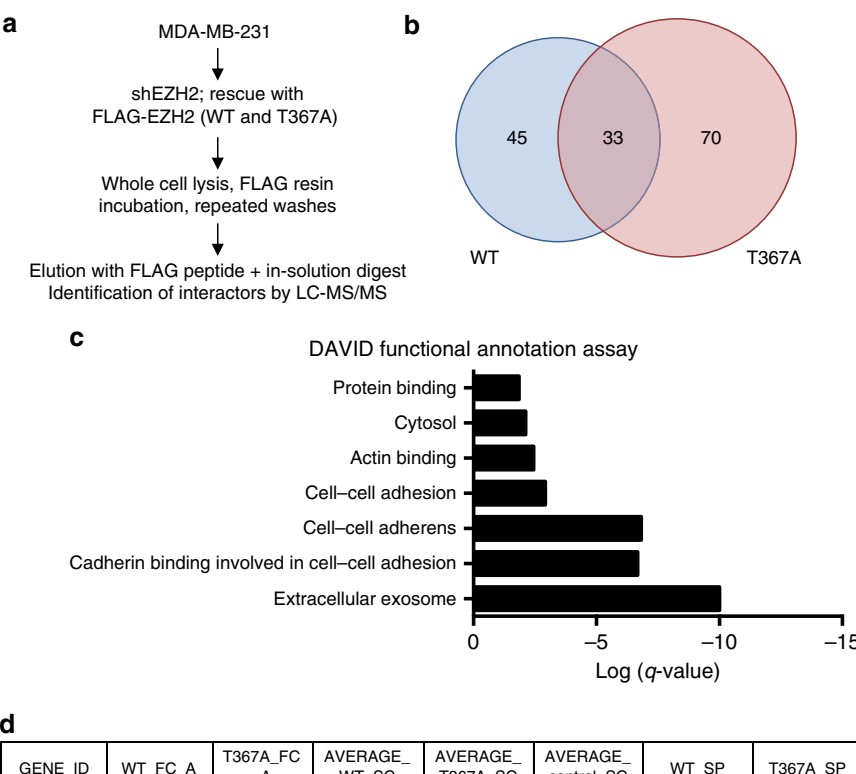

**Fig. 6** The interactome of pEZH2(T367) reveals new cytoplasmic binding proteins. **a** Schematic of mass spectrometry experiment to identify binding partners of EZH2 in MDA-MB-231 cells. Experiment was performed in triplicate. **b** Venn diagram displaying interactors overlap in proteins co-immunoprecipitating with WT- or T367A-EZH2 from the three biological replicates analyzed. **c** DAVID functional annotation analysis of processes enriched in WT-EZH2 over T367A-EZH2. **d** List of differential interactors identified from actin-binding set with fold-change (FC) scores and normalized FC scores based on total EZH2 pulldown. Average WT and T367A spectral counts (SC) and SAINT probabilities (SP) are also tabulated

hypothesis that p38-induced pEZH2(T367)-vinculin interaction may enhance vinculin phosphorylation and activation. We observed that induction of p38 activation increased vinculin Y100 phosphorylation (Fig. 7d), and that WT-EZH2 but not EZH2-T367A rescued phosphorylated vinculin Y100 levels and localization at sites of focal adhesions in MDA-MB-231 cells, suggesting that T367 phosphorylation of EZH2 is required for this function (Fig. 7e, Supplementary Fig. 6F).

Taken together the data suggest that p38-mediated T367 phosphorylation of EZH2 in ER- breast cancer cells promotes a PRC2-independent interaction with cytoplasmic vinculin, leading to phosphorylation of vinculin at Y100 and localization at focal adhesions. Our data document that pEZH2(T367) interacts with cytoplasmic proteins (Supplementary Table 3) in breast cancer cells uncovering a largely unexplored oncogenic mechanism in solid tumors. Figure 7g shows our working model of pEZH2(T367) oncogenic functions.

## Discussion

EZH2 is a bona-fide oncogene in breast cancer, responsible for imparting proliferation, migration, and invasion abilities to breast cancer cells[2,15,20], but the mechanisms are incompletely understood. As the enzymatic component of PRC2, EZH2 has an established transcriptional repression through its catalysis of

histone H3K27 trimethylation. However, the presence of high EZH2 levels in association with low H3K27me3 in aggressive breast cancers suggests that EZH2 operates via a currently unknown H3K27me3-independent mechanism. Here, we discover the presence of upregulated phosphorylated EZH2 at T367 in clinical samples of invasive and metastatic breast carcinoma. Our study shows that p38-mediated phosphorylation at T367 promotes EZH2 cytoplasmic localization and binding to cytoskeletal regulatory proteins and is essential for breast cancer metastasis.

While most studies have focused on the role of EZH2 as a transcriptional repressor in cancer, there is mounting evidence that EZH2 has non-canonical functions involving transcriptional activation and methylation of non-histone proteins. We recently reported that in aggressive ER- breast cancer, EZH2 complexes with RelA/RelB and binds to the *Notch1* promoter to activate transcription independent of its methyltransferase activity[24]. In castration-resistant prostate cancer EZH2 was also found to activate transcription of genes in a methyltransferase-independent manner[25]. EZH2 has also been shown to methylate non-histone substrates; EZH2-mediated methylation of STAT3 leads to STAT3 activation and increased glioblastoma tumorigenicity[26]. The present study reveals a previously undescribed oncogenic mechanism by which p38-mediated EZH2 phosphorylation at T367 promotes breast cancer progression by

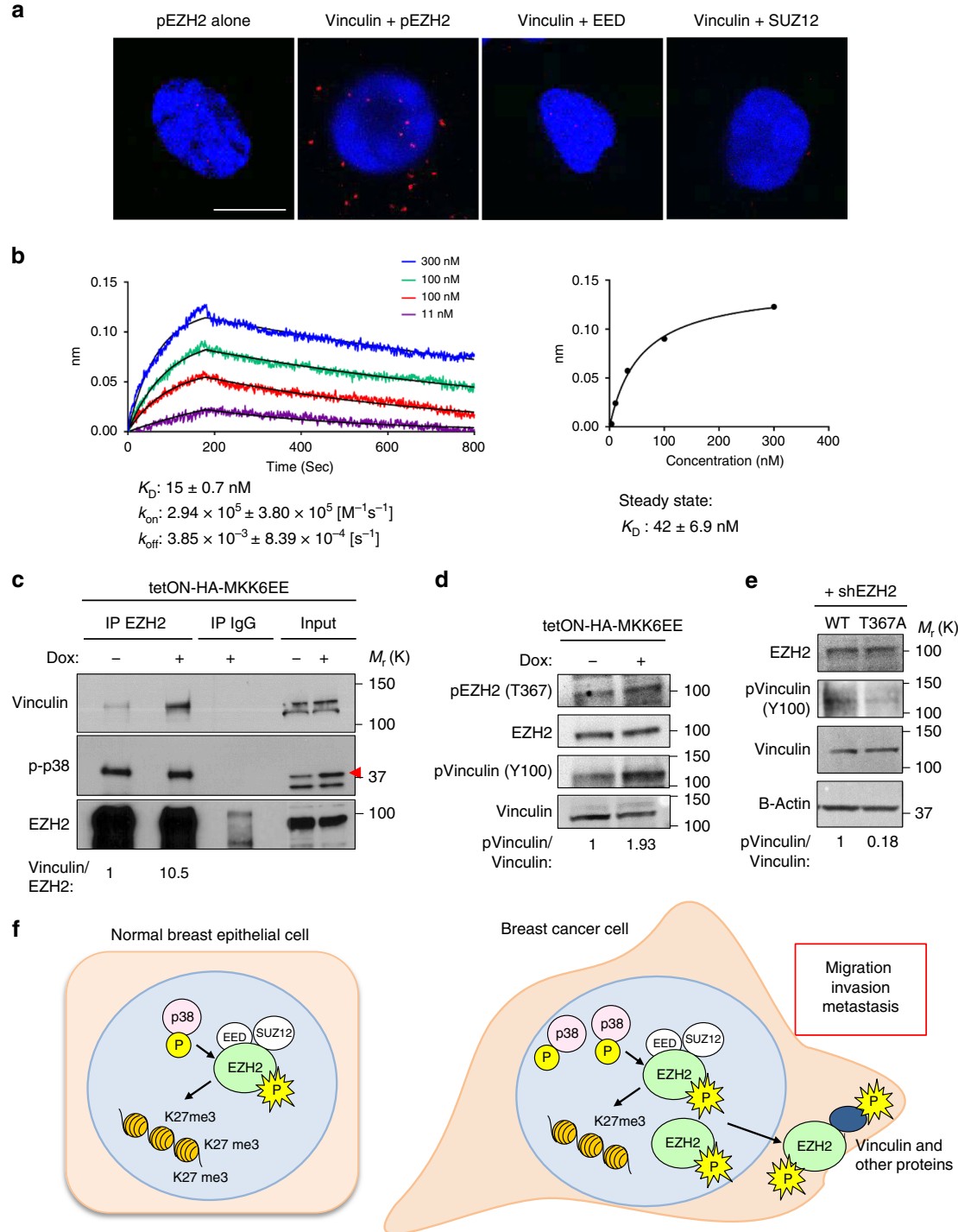

**Fig. 7** EZH2 and vinculin interact in a phosphorylation-dependent manner. **a** Proximity ligation images depicting co-localization with the indicated proteins by red fluorescent dots in MDA-MB-231 cells. Scale bars=10 μm. **b** Quantitative analysis of the direct interaction between recombinant EZH2 and vinculin proteins using BLI. EZH2 was immobilized on a sensor chip, and a concentration series of vinculin protein was added. Sensorgrams and corresponding fitting curves for kinetics constants and affinity determination (left) and corresponding plot of steady state response against concentration (right) for determination of binding affinity. **c** Co-immunoprecipitation experiment from whole cell extracts demonstrating interaction between endogenous EZH2 and vinculin after treatment with doxycycline to induce p38 activation. **d** Western blot of MDA-MB-231 cells transduced with dox-inducible MKK6EE to activate p38α activity. **e** Western blot analysis comparing phospho-vinculin(Y100) in MDA-MB-231 knockdown-rescue WT-EZH2 and T367A-EZH2 cells. **f** Our working model of pEZH2(T367) function in breast tumorigenesis

inducing EZH2 cytoplasmic function and reducing nuclear EZH2 methyltransferase activity on histone H3K27. These conclusions are supported by in vitro and in vivo functional and mechanistic studies and are validated in human breast cancer tissue samples.

Our lab and other investigators have established a role for EZH2 in breast cancer proliferation, migration, and invasion. We have demonstrated that EZH2 shRNA knockdown reduced the size of primary breast cancer xenografts compared to controls[20]. In this study, we show that phosphorylation of EZH2 at

T367 specifically regulates the adhesive, migratory, and invasive properties of breast cancer cells without affecting their proliferation abilities. A phospho-deficient EZH2 mutant promoted proliferation to similar levels of wild type EZH2 but failed to promote breast cancer cell migration, invasion, and adhesion. Demonstrating an essential role for T367 phosphorylation on the ability of breast cancer cells to move and invade, mutation of T367 to alanine resulted in a significant decrease in distant metastatic burden without affecting primary tumor volume, and led to significantly improved metastasis free survival of mice. These findings provide strong evidence for a critical function of pEZH2(T367) in breast cancer metastasis.

In normal breast lobules, when expressed, EZH2 phosphorylated at T367 is localized to the nucleus of epithelial cells. In contrast, 57% of invasive carcinomas and 74% of breast cancer metastasis exhibit upregulated pEZH2 T367 in the cytoplasm of breast cancer cells. Our study revealed that phosphorylation of EZH2 at T367 is sufficient and necessary for cytoplasmic EZH2 localization in breast cancer cells in cell lines and clinical samples of invasive carcinoma. Cytoplasmic EZH2 has been observed previously in murine fibroblasts where it retains methyltransferase activity and regulates actin polymerization[27]. In leukocytes, EZH2 was shown to methylate the cytoplasmic protein talin-1 to enhance migration by inhibiting the binding of talin-1 to F-actin[28]. In breast cancer, we have reported cytoplasmic EZH2 protein in 16% of invasive ER- breast carcinomas from Ghanaian patients[17]. Likewise, cytoplasmic EZH2 expression has been observed in prostate cancer cells[29]. Despite evidence that EZH2 is expressed in the cytoplasm of human malignancies, the function, mechanism, and consequences to cancer progression have remained unexplored. Using EZH2 mutants with a deletion in the nuclear localization signal and a T367 phosphorylation deficient mutant, we directly demonstrate that cytoplasmic localization and T367 phosphorylation are sufficient for EZH2-mediated breast cancer progression.

Through an unbiased proteomics approach, we uncover a phosphorylation-dependent ability of EZH2 to interact with cytoskeletal regulators in breast cancer cells. Among the top pEZH2(T367) actin-binding interactors is vinculin, an F-actin binding protein important for cell-cell and cell-matrix interactions through focal adhesion stabilization[30,31]. Vinculin is over-expressed in human malignancies, including breast cancer, where it regulates cell adhesion and migration[22,23,32–35]. Through several complementary approaches, we find that EZH2 binds with vinculin at high affinity and regulates its activation at Y100 in a T367 phosphorylation-dependent manner. Phosphorylation of vinculin at this site is one of two events critical for cell spreading[36] and cellular transmission of force[37], which likely result from local conformational rearrangements that promote a more open and active state of vinculin[38]. The data suggest one possible mechanism whereby EZH2 regulates this active state through direct interaction. How precisely the interaction of EZH2 and vinculin—which appears to be SUZ12- and EED-independent, and does not involve methylation of vinculin—promotes phosphorylation at this site requires further study. Future studies, such as sizing fractionation and interactive domain mapping will be necessary to determine the biological consequences of this interaction, and to elucidate whether disrupting this interaction if sufficient to abrogate breast cancer cell metastasis.

Our proteomics analyses also uncovered other pEZH2(T367) interactors in breast cancer with roles in cytoskeletal organization that have not been previously studied in this context. Among these interactors are SYNE2, EPS8, EPS8 related protein 2, MLPH, and DBNL. SYNE2 (nesprin-2), an actin-binding nuclear envelope protein that tethers the nucleus to the cytoskeleton, has been shown to promote pancreatic cancer metastasis[39]. EPS8 (Epidermal Growth Factor Receptor Pathway Substrate 8) is a

regulator of actin cytoskeleton dynamics with putative oncogenic functions in breast cancer[40]. EPS8L2 is an EPS8 related protein which links growth factor signaling to actin reorganization[41]. MLPH forms a complex with Rab effector proteins and regulates the movement of melanosomes in the cytoskeleton. DBNL is a cytosolic adaptor protein and putative suppressor of breast cancer metastasis[42]. The contribution of vinculin and other pEZH2 (T367) binding proteins to pEZH2 pro-metastatic functions may lead to new diagnostic, prognostic and therapeutic targets, and warrants in depth further investigations.

Despite interest in phosphorylation as an important non canonical oncogenic mechanism of EZH2 in cancer and the relevance of p38 MAPK in promoting breast cancer metastasis and as a therapeutic target[13], whether p38 phosphorylates EZH2 in breast cancer has not yet been considered. Building on our previous study showing that EZH2 binds to p38α MAPK in ER- breast cancer cells and leads to p38 MAPK signaling activation[15], here we document direct and strong nanomolar binding affinity of EZH2 to p38α using Biolayer Interferometry. Using complementary and independent strategies to induce or inhibit p38α activation, our data show that p38-mediated phosphorylation regulates EZH2 localization, cytoplasmic interacting proteins such as vinculin, and oncogenic function.

In muscle stem cells, injury induced p38-mediated EZH2 T367 phosphorylation leads to enhanced recruitment EZH2 to the Pax7 promoter to promote muscle differentiation and subsequent EZH2 degradation[16,43]. In breast cancer, however, we observe that T367 phosphorylation appears to favor an H3K27me3-independent oncogenic mechanism without significantly affecting EZH2 protein stability. Our data are in agreement with a recent study showing that inhibition of EZH2 phosphorylation at T367 resulted in increased levels of H3K27me3, as well as similarly negligible changes in proliferation with expression of a T367A mutant. However, the authors observed increased migration and invasion with expression of the T367A mutant in MDA-MB-231 and benign MCF12A cells[44]. The discordant functional findings might be explained by the approach used as well as the cellular context; the authors overexpressed wild-type or T367A EZH2 in MDA-MB-231 cells, which express high endogenous levels of EZH2, while we employed a knockdown-rescue approach. The association and mechanistic link between p38-mediated phosphorylation of EZH2 at T367, cytoplasmic localization, and breast cancer progression was validated in vitro, in vivo, and in human breast cancer samples.

Prior to these observations, the existence and role of pEZH2 T367 in the cytoplasm and in the metastatic ability of breast cancer were unknown. Our data show that EZH2 is a phosphorylation substrate of p38 in breast cancer, and that pEZH2(T367) promotes metastasis, at least in part by cytoplasmic localization and interaction with cytoskeletal proteins. Based on our study, targeting this oncogenic mechanism in a subset of ER- breast cancers with high pEZH2(T367) expression using combined EZH2 and p38 inhibition may offer a therapeutic opportunity to interrupt breast cancer metastasis to distant sites. As pEZH2 T367 is also expressed in the cytoplasm of human carcinomas of other organs compared to normal tissues, our findings may shed light into a common mechanism of EZH2 in human cancer.

## Methods

**Cell culture**. Breast cancer cell lines MDA-MB-231, MDA-MB-468, MCF7, and non-tumorigenic breast epithelial cells, MCF10A, were purchased from the American Type Culture Collection and grown under recommended conditions. The CAL51 breast cancer cell line was purchased from German Collection of Microorganism and Cell Cultures (DSMZ GmbH; Cat. No. DSMZ ACC 302) and also grown as recommended. The SUM149 breast cancer cell line was obtained from the laboratory of S. Either (Karmanos Cancer Institute, Detroit) and maintained, as reported previously[45]. Cell lines were authenticated using STR profiling,

and were tested for mycoplasma infection using Sigma LookOut Mycoplasma PCR Detection Kit (Cat MP0035).

Stable knockdown and rescue of EZH2 was achieved by lentiviral transduction of EZH2 with pBabe-myc-EZH2 (wild-type) or pBabe-myc-EZH2 (T367A), both kind gifts from the laboratory of PL Puri[16]. After transduction, cells were selected for antibiotic resistant with 2 μg ml$^{-1}$ puromycin (Sigma Aldrich, #P9620), followed by knockdown using stable short-hairpin interfering RNA (MISSION shRNA, Sigma Aldrich) targeting the 3′UTR of Ezh2 (TRCN0000286227), as previously reported[20]. Oligos in the pLKO.1 vector were packaged into lentiviral particles at the University of Michigan Vector Core.

Inducible activation of p38 MAPK was used using the pBabe pSLIK 3xHA-MKK6-EE neo plasmid, a kind gift from Kevin Janes (Addgene plasmid #47546). Cells were lentivirally transduced and selected with Geneticin (Gibco #10131) and treated with 2 μg ml$^{-1}$ doxycycline (Sigma Aldrich, #D3072) to induce activation of MKK6 as previously reported[16]. Cells were treated The p38 inhibitors SB202190 (Cell Signaling #8158) and SB203580 (Cell Signaling #5633) and were used at 20 μM for 48 h as previously reported[15] p38a knockdown was achieved using stable short-hairpin interfering RNA (MISSION shRNA, Sigma Aldrich) (TRCN0000000510). Primers used for mutagenesis reactions to create FLAG-T367A constructs from FLAG-EZH2, as well as primers used for Sanger sequencing of EZH2 constructs can be found in Supplementary Table 4.

**Western blotting and immunoprecipitations**. Western blotting analyses were carried out as previously reported using 50 μg of whole cell extract, as previously reported[20]. Briefly, cells were lysed in RIPA lysis buffer (Pierce #89900) with protease and phosphatase inhibitors (Thermo Scientific #1861281). Samples were resolved by SDS–PAGE, transferred onto PVDF membranes, and membranes were blocked and incubated with primary antibodies in 5% BSA (Sigma Aldrich, #A3059) in TBS-T (Bio-Rad, #161-0372 with 0.05% Tween 20) or 5% milk (Bio-Rad #170-6404) in TBS-T at 4 °C overnight. Protein signals were detected using enhanced chemiluminescence (Pierce, #32106) as per the manufacturer's instructions. Primary antibodies used included Cell Signaling antibodies: EZH2 (#5246), Histone H3 (#9715), myc-tag (#2276), p38α MAPK (9218), trimethyl-histone H3 (Lys27) (#9733), SP1 (#9389) SUZ12 (#3737), phospho-p38 MAPK (#4511) phospho-Hsp27 (#2401); Abcam antibodies: EED (#ab4469), vinculin (#ab18058); Thermo Antibody: phospho-vinculin Y100 (Catalog #44-1074 G); B-Actin HRP (Santa Cruz, #sc47778) was used as for loading control. Secondary antibodies used were Amersham ECL anti-rabbit IgG HRP-linked (GE Healthcare Life Sciences, #NA934) or Amersham ECL anti-mouse IgG HRP-linked (GE Healthcare Life Sciences, #NA931). Uncropped images of western blots used in this article can be found in Supplementary Figure 7.

Immunoprecipitations of endogenous proteins was performed using magnetic Dynabeads following protocol instructions (Invitrogen, #10007D). Briefly, cells were lysed in IP lysis buffer (Pierce #87788) with protease and phosphatase inhibitors (Thermo Scientific #1861281). Dynabeads were washed and incubated for 10 min with rotation with supplied antibody-washing buffer containing antibody for bead-antibody conjugation. Antibodies used for immunoprecipitation included EZH2 (Cell Signaling #5246), pEZH2 (custom antibody), and vinculin (#ab18058). After conjugation, beads were washed with supplied antibody-washing buffer and incubated with protein extract overnight at 4 °C. The next day, dynabead-antibody-antigen complexes were washed in stringent conditions and eluted with SDS-Laemmli Sample Buffer. Immunoprecipitations of myc-tagged proteins were performed using anti c-myc agarose resin (Pierce #20168) following the manufacturer's instructions.

For fractionation, the Thermo Subcellular Fractionation Kit for Cultured Cells (Catalog 78840) was used.

**Wound healing, invasion, microfluidic migration, and cell attachment assays**. Wound healing assays were performed by seeding cells in complete media a 6-well plate for 24–48 h until a confluent monolayer had formed. Linear scratches were made using a sterile 200 μl pipette tip. Monolayers were washed three times with PBS to remove detached cells, and then complete media was added. Photographs of the wound were taken immediately after wound formation and 24 h after with phase contrast microscopy. Wound area was measured over time using ImageJ.

In vitro invasion assays were performed using a 24-well matrigel invasion chamber (BD Biosciences, #354480), per manufacturer's instructions. All invasion experiments were performed with technical triplicates, and repeated at least three times with biological replicates. Cells that had invaded through the matrigel membrane were fixed with methanol, stained with crystal violet, photographed at high resolution, and counted manually using ImageJ. The represetnative whole inserts were imaged under the same conditions, and are shown in this paper after increasing brightness by 20% across all images in Microsoft Powerpoint.

Microfluidic migration assays were performed using a previously published microfluidic migration platform[46,47]. To achieve higher throughput, the design was modified to have 450 migration channels per device, and the migration channel was designed to be 5 μm in height, 30 μm in width, and 1 mm in length. Before cell loading, collagen solution (1.45 mL Collagen (Collagen Type 1, 354236, BD Biosciences) and 0.1 mL acetic acid in 50 mL DI Water) was used to prime the device for 1 h, and the cell culture medium flowed through the channel for 1 h for better cell adhesion and viability. The cells were trypsinized, centrifuged, and then

re-suspended to a concentration of $4 \times 10^5$ cells ml$^{-1}$ for loading into the device. After cell loading, the cell suspension in the inlet was replaced by serum-free cell culture media, and 10% FBS serum cell culture media was applied to the other inlet to induce chemotactic migration. The microfluidic chip was then put into an incubator, and migration distance was measured based on the final cell position after 24 h of incubation without medium replenishment. For data collection, cells were stained by LIVE/DEAD® Viability/Cytotoxicity Kit (Invitrogen, L3224) to distinguish live and dead cells. To have consistent results, we only use the data from central 300 (out of 450) migration channels. The images were analyzed by custom MATLAB code automatically[48]. Cells were identified based on their fluorescence, and debris was ignored by their small size. For all conditions in this work, 4 replicates (a total of 1200 channels) were performed. Box graphs were plotted using Origin 9.0. The bottom and top of the box are the first and third quartiles, and the band inside the box is always the second quartile (the median). The ends of the whiskers represent the 5th percentile and the 95th percentile. The square inside the box indicates the mean, and the x outside the box indicates the minimum and maximum of all of the data.

Cell attachment assays were performed by trypsinizing 70% confluent cell dishes and seeding $1 \times 10^5$ cells in a 12 well plate. After 30 min, non-adherent cells were removed by washing wells with PBS three times. Adherent cells were then imaged and entire wells were counted using ImageJ.

**Determination of the binding affinity using Bio-Layer Interferometry (BLI) technology**. Recombinant EZH2 (GST-EZH2 aa 2-end; MW = 114 kDa; Bioscience) protein was biotinylated using the Thermo EZ-link Sulpho-NHS-LC-biotin biotinylation kit (cat. 21435). EZH2 protein and biotin were mixed in a 1:1 molar ratio in HBS buffer (10 mM HEPES pH 7.4, 150 mM NaCl) on ice for 2 h. Reaction mixture was dialyzed in HBS buffer to remove excess biotin.

BLI experiments were performed using an OctetRED96 instrument from PALL/ForteBio. All assays were run at 30 °C using HBS-P buffer (10 mM HEPES pH 7.4, 150 mM NaCl, 0.005% tween-20) with continuous 1000 rpm shaking. Biotinylated EZH2 was immobilized on Super Streptavidin (SSA) biosensors (ForteBio) by dipping sensors in 20 μg ml$^{-1}$ protein solutions. Biotin labeled streptavidin protein was immobilized on SSA sensors and used as inactive reference controls. Recombinant p38α (His-p38α aa 1–360; MW = 43 kDa; Abcam) allowed to associate for 2 min and dissociate for 2 min. Collected raw kinetic data collected were processed with the Data Analysis software provide by ForteBio using double referencing in which both the buffer only sensors and inactive protein sensors were subtracted. Resulting data were analyzed based on the 1:1 binding model and kinetic parameters $k_{on}$, $k_{off}$ and $K_d$ were determined as well as steady state binding affinity.

**Spontaneous metastasis model and xenograft immunohistochemistry**. Eight-week old severe combined immunodeficiency mice (Jackson Laboratories) were used for examining tumorigenicity, as previously reported[15]. Briefly, GFP-Firefly-luciferase expressing MDA-MB-231 shVector, shEZH2 + pBabe, shEZH2+WT-EZH2 or shEZH2+T367A-EZH2 cells were orthotopically injected into the right inguinal mammary fat pad of anesthetized mice at a concentration of $10^6$ cells resuspended in 50 μl of matrigel ($n = 10$ mice per group). Primary tumor growth was monitored semiweekly by caliper measurement as strong BLI signals quickly become saturated by rapid growth of primary MDA-MB-231 tumors. Metastases were monitored using bioluminescence imaging as previously described. Briefly, mice were anesthetized and injected i.p. with 75 mg/kg D-Luciferin (Xenogen) resuspended in PBS. Bioluminescence images were acquired using the IVIS imaging system (Xenogen) within ~2–5 min after injection. Analysis was performed using the Living Image software platform (Xenogen) by measuring photon flux, measured in photons/s/cm$^2$/sr, by using a region of interest (ROI) drawn around the biolu-minescence signal to be measured and subtracting background measurements. All mice were sacrificed when the first primary tumor size reached 2 cm$^3$.

**Immunohistochemistry, immunofluorescence, and proximity-ligation assays**. Tumors, lungs, and bones of mice were collected, fixed in 10% neutral buffered formalin, and embedded in paraffin for immunohistochemistry. Immuno-fluorescence was done by seeding cells into 2-well chambered slides (Thermo Fisher Lab-Tek #154461). After 24 h of seeding, cells were fixed with 4% PFA diluted in PBS for 15 min at room temperature, rinsed three times with PBS, and blocked for 1 h using blocking buffer, 5% normal goat serum containing 0.3% Triton X-100 in PBS. After blocking, slides were incubated with primary antibody diluted in antibody buffer (5% bovine serum albumin containing 0.3% Triton X-100 in PBS) at 4 °C overnight. Next day, slides were washed three times with PBS and incubated with fluorescent secondary antibodies (Alexafluor goat-anti Rabbit 488 Cat # A-11008 or goat anti-mouse 594 Cat A-11005). Slides were washed three times with PBS and coverslipped using ProLong Diamond Antifade Mountant with DAPI (Thermo Fisher, Cat# P36962). Slides were imaged using Leica SP5 Inverted 2-Photon FLIM Confocal, and image analysis was performed using ImageJ. For immunofluorescence imaging and quantitation of phospho-vinculin Y100 focal adhesion, 8-well chambered slides slides (Thermo Fisher Lab-Tek Cat #154534) were first coated with fibronectin (Sigma Fibronectin F0895) per the manufacturer's coating protocol at a dilution of 2 μg ml$^{-1}$, and immunofluorescence was carried out as outlined above. For post-imaging analysis, we followed a previously pub-lished protocol from Horzum et al.[49], which details a step-by-step quantitative

analysis of focal adhesions from MDA-MB-231 breast cancer cells, quantifying focal adhesions from 70 to 100 cells per condition across at least 5 separate fields.

For proximity ligation assays, cells were seeded in 8-well chambered slides (Thermo Fisher Lab-Tek Cat #154534). After 24 h of seeding, cells were fixed with 4% PFA diluted in PBS for 15 min at room temperature, rinsed three times with PBS, and blocked for 1 h using blocking buffer, 5% normal goat serum containing 0.3% Triton X-100 in PBS. After blocking, slides were incubated with primary antibody diluted in antibody buffer (5% bovine serum albumin containing 0.3% Triton X-100 in PBS) at 4 °C overnight. Next day, slides were washed three times with PBS and incubated with DuoLink PLA probes (Sigma, Cat #DUO92101) and protocol for PLA secondary antibody incubation, ligation, amplification, and washes were performed following the manufacturer's protocol. Slides were imaged using Leica SP5 Inverted 2-Photon FLIM Confocal. Positive signals were normalized to single-primary antibody control (pEZH2, 1:500) and image analysis was performed using ImageJ. Images were taken under the same conditions, and if manipulated for representative images, brightness was increased across all images equally.

**Affinity-purification mass spectrometry.** MDA-MB-231 knockdown rescue cells expressing FLAG-EZH2 or FLAG-T367A were washed three times with PBS, harvested, and lysed in Pierce IP Lysis Buffer (#87797) containing protease and phosphatase inhibitors (Thermo Scientific #1861281) and immunoprecipitated with anti-FLAG antibody beads (Sigma M8823). On-bead digestion followed by LC–MS/MS analysis was performed following the protocol optimized at the Proteomics Resource Facility at the University of Michigan. Briefly, the beads were resuspended in 50 ml of 100 mM ammonium bicarbonate buffer (pH ~8). Upon reduction (10 mM DTT) and alkylation (65 mM 2-chloroacetamide) of the cysteines, proteins were digested with 500 ng of sequencing grade, modified trypsin (Promega). Resulting peptides were resolved on a nano-capillary reverse phase column (Acclaim PepMap C18, 2 μm, 50 cm, ThermoScientific) using 0.1% formic acid/acetonitrile gradient at 300 nl min$^{-1}$ (2–25% acetonitrile in 105 min; 25–40% in 20 min, followed by a 90% acetonitrile wash for 10 min and a further 25 min re-equilibration with 2% acetonitrile) and directly introduced in to Q Exactive HF mass spectrometer (Thermo Scientific, San Jose CA). MS1 scans were acquired at 120 K resolution. Data-dependent high-energy C-trap dissociation MS/MS spectra were acquired with top speed option (3 s) following each MS1 scan (relative CE ~28%). Proteins were identified by searching the data against *Homo sapiens* database (UniProtKB, v2014-4-13) using Proteome Discoverer (v2.1, Thermo Scientific). Search parameters included MS1 mass tolerance of 10 ppm and fragment tolerance of 0.1 Da; two missed cleavages were allowed; carbamidimethylation of cysteine was considered fixed modification and oxidation of methionine; deamidation of aspergine and glutamine; phosphorylation of Serine, Threonine and Tyrosine were considered as variable modifications. Percolator algorithm was used for discriminating between correct and incorrect identification and peptides/proteins with <1% FDR (false discovery rate) were retained for further analysis.

Interactions with EZH2 and mutant EZH2 were scored using empirical fold-change scores (FC) and significance analysis of interactome (SAINT) probabilities for each interaction calculated using the CRAPome resource[21]. To calculate the FC scores (the primary FC-A score) and SAINT probabilities (using SAINTexpress[50]), the three FLAG-IP replicates of cells expressing only the empty vector were used as negative controls. Replicates were combined in FC scoring and in SAINT probability calculation using average values of the three biological replicates. Briefly, the FC scores represent the increase (or decrease) in protein abundances (estimated using MS/MS spectral counts) in bait IPs relative to the control samples. SAINT calculates the probability that an interaction is a true positive using a model where true-positive and false-positive interactions for each bait are modeled statistically as distinct Poisson distributions. A value of 1 indicates a high probability of a bona-fide interaction[51].

PANTHER cellular component (gene ontology) analyses were performed on protein sets that were filtered using a SAINT probability cutoff of ≥0.7 and an FC score of ≥2. DAVID functional analysis was performed using a stricter SAINT cutoff of 0.9. (https://david.ncifcrf.gov/content.jsp?file=citation.htm; http://pantherdb.org/citePanther.jsp). The background for these analyses were set as all of the proteins identified in the LC-MS/MS experiment (5800 proteins).

Venn diagrams were generated using the VennDiagram package in R version 3.3.2. Data-framing was performed using RStudio version 0.98. Graphs of enrichment analyses were generated in Prism6.

**Tissue samples and immunohistochemistry.** Tissues from 104 invasive carcinomas arranged in triplicate samples in a high density tissue microarray (TMA), 19 normal breast tissues, and 23 tissue samples of distant metastasis, previously characterized by our group were employed[52–54] (obtained with University of Michigan Institutional Review Board approval No. HUM00050330). In total 5 μm-thick paraffin-embedded sections were de-paraffinized in xylene and rehydrated through graded alcohols to water. Heat Induced Epitope Retrieval (HIER) was performed in the Decloaking Chamber (Biocare Medical) with Target Retrieval, pH 6.0 (DakoCytomation). Slides were incubated in 3% hydrogen peroxide for 5 min to quench endogenous peroxidases. Anti-pEZH2(T367) (1:8000) developed by our lab and anti-H3K27me3 (Cell Signaling Tri-Methyl-Histone H3 (Lys27) (C36B11) Rabbit mAb #9733, 1:200) were incubated with the tissue sections for 1.5 h at room

temperature. Antibodies were detected with Envision+HRP Labeled Polymer (DakoCytomation) for 30 min at room temperature. HRP staining was visualized with the DAB+Kit (DakoCytomation). Negative control slides were run. Slides were counterstained in hematoxylin, blued in running tap water, dehydrated through graded alcohols, cleared in xylene and then mounted with Permount. Expression of pEZH2 (T367) and H3K27me3 was analyzed blindly by two observers, at least twice. pEZH2 (T367) staining was categorized as nuclear or cytoplasmic, and as high and low based on the presence or absence of protein expression. The expression of H3K27me3 in the nucleus of cancer cells was scored using a four-tiered system based on intensity of staining and percentage of staining cells, with scores 1–2 categorized as low, and 3–4 as high[52,55]

**Statistics.** Results are presented as mean±SD or mean±SEM, unless otherwise noted. Comparisons between two groups were performed using an unpaired two-sided Student's *t* test for continuous variables or Chi-square test for categorical variables. These analyses were performed using GraphPad Prism software for all in vitro and in vivo studies. A *p* value of <0.05 was considered statistically significant.

**Study approval.** All procedures using animals were conducted in accordance with the NIH Guide for the Care and Use of Laboratory Animals and were approved by the Institutional Animal Care and Use Committee at the University of Michigan (UCUCA#PRO 00005009).

**Data availibility.** The mass spectrometry proteomics data have been deposited to the ProteomeXchange Consortium via the PRIDE[56] partner repository with the dataset identifier PXD010073. Other data that support the findings of this study are available from the corresponding author upon reasonable request.

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

## Acknowledgements

We thank members of the Kleer lab for discussions during the execution of this project. We thank the lab of Dr. Pier Lorenzo Puri for myc-EZH2 and T367A-EZH2 vectors. This work was supported by the National institutes of Health (NIH) grants R01CA125577 and R01CA107469 (C.G.K.), F30CA19084 (T.A.), R25GM086262 (PREP program, C.A.-G.), R01GM094231 (A.I.N.), T32CA140044 (J.R), the University of Michigan Rogel Cancer Center support grant P30CA046592, the Karlene Kulp Fund Judy & Ken Robinson Fund (C.G.K.) and the Department of Defense award W81XWH-15-1-0019 (C.G.K. and Z.N.-C.).

## Author contributions

T.A. designed and performed experiments, analyzed data, and wrote the paper. C.A.-G. assisted with experimental execution of immunoprecipitations. J.R. performed analysis of LC MS/MS data. Y.-C.C. developed, performed, and analyzed high throughput single cell migration assays. H.S.K. contributed with in vitro experimental data. E.Y. contributed with microfluidics migration assays. S.G. performed bio-layer interferometry. V.B. and A.I.N. performed and assisted with LC MS/MS data interpretation. A.M. aided with analysis of LC MS/MS data and experimental strategies. M.E.G assisted with project design, experimental strategies, and data analysis. K.M.K. performed statistical analyses on human tissues. Z.N.-C. contributed with bio-layer interferometry data analysis and experimental strategies. C.G.K. conceived the study, contributed to experimental design, and wrote paper.

## Additional information

**Competing interests:** The authors declare no competing interests.

