## [Peer Review File · Nature Communications]

Reviewers' comments:

Reviewer #1 (Remarks to the Author):

The author's main claim in this paper is that EZH2 T367 phosphorylation induces its localization to cytoplasm and promote breast cancer metastasis partly by interacting with cytoskeletal regulator such as vinculin. However, their data requires few more experiments to support their major claim. Below are major points, which need clarification, before consideration for publication in Nature communication.

Major points

1. Authors made homemade EZH2 T367 phosphorylation specific antibody and performed immunohistochemical analysis to show phosphorylated EZH2 (T367) is present in cytoplasm. Also, they showed the cytoplasmic fraction of EZH2 is reduced when Threonine 367 is mutated to alanine which is phosphorylation deficient mutant. This is one of the important observations to support their major claim, therefore, they should confirm this by other methods such as Western blotting. They should fractionate nucleus and cytoplasm and check the enrichment of phosphorylated EZH2 T367 in cytoplasmic fraction.
2. In figure 4 and 5, they showed that rescue of EZH2 T367A in EZH2 knockdown breast cancer cells reduced breast cancer migration, invasion, cell adhesion, and metastasis, suggesting phosphorylation of EZH2 T367 is required. They should also rescue with EZH2 T367D that mimics phosphorylation and check the migration, invasion, cell adhesion and metastasis of breast cancer cells are increased compared to WT breast cancer cells.
3. Although the interaction between EZH2 and vinculin was shown by proximity ligation imaging and immunoprecipitation, the mechanism of how EZH2 interaction with vinculin enhances cell migration is not known. The mechanistic insight is missing.
4. Author should validate their EZH2 T367 phosphorylation specific antibody by checking the band disappears in EZH2 T367A rescued cell line.

Reviewer #2 (Remarks to the Author):

Anwar et al. p38-mediated phosphorylationbreast cancer metastasis

In this study, the authors have focused on the non-canonical function of EZH2 in ER-negative breast cancer. They developed and validated a polyclonal antibody with specificity to pEZH2 (T367), and using TMAs showed that pEZH2 is expressed in the cytoplasm of most invasive and metastatic breast cancer. p38-mediated phosphorylation of T367A in EZH2 was proposed to mediate invasion/migration in vitro and metastasis in vivo by interacting with cytoskeletal regulators in the cytoplasm. The topic of metastasis and the mechanistic role of EZH2 are important and timely. However, deficiencies exist including lack of conceptual advancements as previous studies have already demonstrated cytoplasmic EZH2, interaction of p38 and EZH2 and p38 mediating EZH2 phosphorylation. Furthermore, in this study overexpression of key transgenes may result in non-physiological effects, and lead to over-interpretation of the data. Many in vitro and all in vivo experiments rely on one breast cancer cell line, without validation in another model. Most importantly, the mechanistic angle, which is the novel component of this study, remains underdeveloped.

Major Points:

- 1) Figure 1 relies on data from breast cancer TMA. However, this reviewer had a difficult time to

evaluate this data due to lack of information associated with this microarray, patient characteristics, scoring methods used, data analysis etc.

2) Stable 3'UTR EZH2 knockdown mediated loss of invasion/migration was shown to be rescued by adenovirus expressing full length (WT-EZH2) and EZH2 Δ NLS mutant. However, western blot in Sup Fig. 3b shows that WT-EZH2 was unable to rescue H3K27me₃, casting doubts on the rescue of invasive and migratory phenotype shown in Fig 3E-F.

3) Is Δ NLS functional? Lack of evidence showing that Δ NLS incorporates into the PRC2 complex.

4) The use of EZH2 T367A to delineate the role of pEZH2 in Fig 4-5 is appreciated. However, overexpression of EZH2 could cause non-physiological effects, and therefore, it may be better to use cells with mutation in endogenous EZH2.

5) There are some issues with in vivo experiments showing primary tumors and metastatic lesions in Fig 5. As the primary tumors were not resected, the authors need to ensure that there is no bleeding of primary tumor signals into the lungs. Is the size of the primary tumor impacting the BLI signal of the metastatic lesions?

6) In Fig 5B, how was metastasis free survival determined without confounding effects of large primary tumors in shvector, wt-EZH2 and T367A cohorts.

7) Fig 5d. Why is there such high variability in metastasis within control shVector.

8) The primary tumor growth and metastasis assays with MDA-231 is >6-8 weeks. It is important to show that Adenoviral infected cells continue to express transgenes during this window.

9) The authors show that phosphorylated T367 EZH2 interacts with vinculin, however the mechanistic aspects are not sufficiently explored. The downstream mediators of phosphorylated T367 EZH2 need to be elucidated. For example, where does EZH2 bind vinculin? Does EZH2 impact vinculin activation, binding partners or localization? Given that pEZH2 incorporates in PRC2, why are PRC2 components missing in Fig 7b. Is the invasive phenotype of phosphorylated T367 EZH2 due to its interaction with vinculin? How does this impact invasive behavior of MDA-MB-231 cells in vitro and metastasis in vivo?

10) It is important to determine the relationship between pEZH2 and H3K27me₃ and how does this impact carcinogenesis. Also in this context, its important to discuss the authors previous findings that EZH2 inhibition impacts p38 signaling.

Other points

Ref #6 does not show that EZH2 and low H3K27me₃ correlate with poor prognosis of ER- breast cancers, as the authors state.

Ref # 16 Placios et al does not show that p38 contributes to T367 phosphorylation as the authors state. It shows that T372 phosphorylation is impacted.

Please quantify western blot data in Fig 1C.

For functional analysis, SUM159 will be used as it expressed increased pEZH2. However, data in Fig 1c is from SUM149.

There is some discussion in the literature regarding specificity issues with p38 inhibitors including the ones used in this study.

In Fig 4A, Ad EZH2 and T367A show higher molecular weight compared to Wt EZH2. Myc Tag alone is unlikely to make such a difference around 100kD.

REVIEWER 1

1. *“Authors made homemade EZH2 T367 phosphorylation specific antibody and performed immunohistochemical analysis to show phosphorylated EZH2 (T367) is present in cytoplasm. Also, they showed the cytoplasmic fraction of EZH2 is reduced when Threonine 367 is mutated to alanine which is phosphorylation deficient mutant. This is one of the important observations to support their major claim, therefore, they should confirm this by other methods such as Western blotting. They should fractionate nucleus and cytoplasm and check the enrichment of phosphorylated EZH2 T367 in cytoplasmic fraction.”*

As the reviewer suggested, we have performed multiple fractionation experiments using MDA-MB-231 and -468 triple negative breast cancer cells, which demonstrate enrichment of phosphorylated EZH2 T367 in the cytoplasmic-enriched fraction. Please see **new Suppl. Fig. 3b**.

2. *“In figure 4 and 5, they showed that rescue of EZH2 T367A in EZH2 knockdown breast cancer cells reduced breast cancer migration, invasion, cell adhesion, and metastasis, suggesting phosphorylation of EZH2 T367 is required. They should also rescue with EZH2 T367D that mimics phosphorylation and check the migration, invasion, cell adhesion and metastasis of breast cancer cells are increased compared to WT breast cancer cells.”*

We understand the reviewer. Studies have shown that aspartic and glutamic acid often fail to fully recapitulate the effects of phosphorylation, particularly in facilitating adaptor interactions, due to differences in size, geometry, and degree of negative charge (Chen and Cole, 2015). For example, a phosphoserine, but not a phosphomimetic protein, can compete for binding with an adjacent lysine, which results in partial unfolding and promotes new protein-protein interactions (Skinner et al., 2017). In an effort to address the reviewer’s comment, we generated EZH2-T367D and EZH2-T367E phosphomimetic constructs and performed knockdown-rescue tests with two cell lines. Despite these efforts, we found no effects of the phosphomimetic mutants in the migration and invasion abilities of MDA-MB-231 and MDA-MB-468 breast cancer cells. To strengthen our results using EZH2 T367A, in addition to MDA-MB-231 cell, we have performed functional experiments in MDA-MB-468 and SUM159 breast cell lines with at least 3 independent biological replicates for each cell line (**new Suppl. Fig. 4a-d**). These new data further validate the critical role of T367 phosphorylation in breast cancer migration and invasion.

3. *“Although the interaction between EZH2 and vinculin was shown by proximity ligation imaging and immunoprecipitation, the mechanism of how EZH2 interaction with vinculin enhances cell migration is not known.”*

We are grateful for the reviewer’s comment which led us to make several new observations. Our new quantitative Bio-Layer Interferometry studies showed a direct binding between EZH2 and vinculin with strong affinity ($k_D 15 \pm 0.7\text{nM}$). Further, we discovered that p38-mediated EZH2 phosphorylation potentiates cytoplasmic pEZH2-vinculin binding, promoting vinculin phosphorylation at Y100 and increasing p-vinculin (Y100) localization at focal adhesions. Importantly, we found that pEZH2-vinculin interaction is independent of PRC2. This new mechanism was demonstrated using complementary and independent approaches and multiple breast cancer cell lines.

We have performed new mechanistic studies, which show that activation of p38 using a tetON-HA-MKK6EE construct induced a 10-fold increase in pEZH2-vinculin binding compared to controls, and was sufficient to increase p-vinculin(Y100) (**Figure 7c and new Figure 7d**). Using our knockdown-rescue system, we demonstrate that the EZH2 phosphorylation deficient mutant, T367A, does not rescue p-vinculin levels leading to reduced p-vinculin at sites of focal adhesions compared to EZH2-WT, demonstrating a requirement for pEZH2(T367) for this function (**new Fig. 7e and new Suppl. Fig. 6f**).

Our revised paper also investigates in depth the EZH2-vinculin interaction in vivo and in vitro. In addition to co-immunoprecipitation, immunofluorescence, and proximity ligation assay (PLA) studies in breast cancer cells, we have performed quantitative Bio-Layer Interferometry using recombinant proteins, which demonstrates direct binding with strong affinity between these proteins, which was previously unknown. These data have been added to **new Fig. 7b**.

Based on the methyltransferase function of EZH2 and our data that pEZH2(T367) associates with PRC2 members (**Suppl. Fig. 5b and 6e**), we investigated whether pEZH2(T367)-vinculin interaction resulted in vinculin methylation. In our studies using methyltransferase assays and mass spectrometry analyses after incubation with PRC2, we were unable to detect vinculin methylation (not shown). These data are further supported by PLA showing that vinculin does not interact with PRC2 members SUZ12 and EED (**Fig. 7a**). The ability of pEZH2(T367) to incorporate PRC2 suggests the novel hypothesis that pEZH2 may methylate cytoplasmic proteins, which we will investigate in future studies.

4. *“Author should validate their EZH2 T367 phosphorylation specific antibody by checking the band disappears in EZH2 T367A rescued cell line.”*

As suggested by the reviewer, we have now further validated our pEZH2(T367) specific antibody, and have added the data to **new Suppl. Fig. 1f**.

REVIEWER 2

General remarks: *“In this study, the authors have focused on the non-canonical function of EZH2 in ER-negative breast cancer. They developed and validated a polyclonal antibody with specificity to pEZH2 (T367), and using TMAs showed that pEZH2 is expressed in the cytoplasm of most invasive and metastatic breast cancer. p38-mediated phosphorylation of T367A in EZH2 was proposed to mediate invasion/migration in vitro and metastasis in vivo by interacting with cytoskeletal regulators in the cytoplasm. The topic of metastasis and the mechanistic role of EZH2 are important and timely.”*

We thank the reviewer for the positive comments on the significance of the work for breast cancer metastasis.

1. *“Figure 1 relies on data from breast cancer TMA. However, this reviewer had a difficult time to evaluate this data due to lack of information associated with this microarray, patient characteristics, scoring methods used, data analysis etc.”*

We have provided further details on the human tissues, characteristics of the patients, as well as details on the quantification of the immunohistochemistry in the supplementary methods and in **new Suppl. Tables 1 and 2**. The statistical analyses were performed by Kelley Kidwell, Research Associate Professor of Biostatistics and collaborator in this study.

2. *“Stable 3'UTR EZH2 knockdown mediated loss of invasion/migration was shown to be rescued by adenovirus expressing full length (WT-EZH2) and EZH2ΔNLS mutant. However, western blot in Sup Fig. 3b shows that WT-EZH2 was unable to rescue H3K27me3, casting doubts on the rescue of invasive and migratory phenotype shown in Fig 3E-F.”*

We understand the reviewer. We have repeated this experiment multiple times and show that EZH2 knockdown reduces H3K27me3 to 17% of the control levels, WT-EZH2 effectively rescues H3K27me3 to 37% of the control levels, and rescue with EZH2 Δ NLS is unable to rescue H3K27me3 (9% of the control levels) supporting the functional studies. We have added the quantitation to **Suppl. Fig 3c**.

3. *“Is Δ NLS functional? Lack of evidence showing that Δ NLS incorporates into the PRC2 complex.”*

We appreciate the reviewer’s question, which led us to investigate in detail whether Δ NLS mutant has the capacity to interact with PRC2 members. New data from co-immunoprecipitation experiments using whole cell lysates provide evidence that Δ NLS-EZH2 is able to interact with PRC2 core members SUZ12 and EED in ER negative human breast cancer cell lines (**new Suppl. Fig 3d**). These data suggest the hypothesis that EZH2 may function in the cytoplasm as part of PRC2, which will be investigated in future studies. The data are in line with and supported by observations of previous studies that show that Δ NLS EZH2 isolated from HEK293T cells maintains its methyltransferase activity in vitro (Su et al., 2005).

4. *“The use of EZH2 T367A to delineate the role of pEZH2 in Fig 4-5 is appreciated. However, overexpression of EZH2 could cause non-physiological effects, and therefore, it may be better to use cells with mutation in endogenous EZH2.”*

We understand the concern that there is a possible effect of the lower endogenous levels of EZH2 remaining after lentivirus-mediated EZH2 knockdown. Western blots show a similar effective EZH2 KD (approx. 80%) in the 3 experimental groups (EZH2 KD, and the rescue lines with WT-EZH2 and T367A-EZH2) (**Fig 4a**). We performed titration experiments for the viral transductions to achieve similar levels of overexpression of the WT-EZH2 and T367A-EZH2 in the rescue cell lines (Western blots in Fig 4a). We have also performed this experiment in other cell lines (SUM159 and MDA-MB-468, **new Suppl. Fig. 4a**), and obtained similar results. Taken together, these results support the functional in vitro and in vivo studies, and the significant functional differences between the experimental groups.

5. *“Fig 5. As the primary tumors were not resected, the authors need to ensure that there is no bleeding of primary tumor signals into the lungs. Is the size of the primary tumor impacting the BLI signal of the metastatic lesions? How was metastasis free survival determined without confounding effects of large primary tumors in shvector, wt-EZH2 and T367A cohorts. In Fig 5d, why is there such high variability in metastasis within control shVector?”*

We understand the reviewer, and have experience with the challenges of BLI. To determine the size of the metastatic focus as accurately as possible, we have carefully shielded the primary tumors to measure BLI signal of the metastatic lesion, as performed by other investigators and in our own laboratory (Pal et al., 2012; Tosatto et al., 2016). The variability in metastasis in the control shVector may reflect the characteristics inherent of the breast cancer cell lines, and has been observed by other investigators (e.g. (Zhang et al., 2013)). We have revised Fig. 5 to more clearly show that rescue with T367A-EZH2 promotes primary tumor growth similar to WT-EZH2 (**new Fig. 5 a-b**), but that rescue with T367A-EZH2 significantly reduces metastatic dissemination and prolongs survival compared to WT-EZH2 (**Fig. 5c-e**).

6. *“The primary tumor growth and metastasis assays with MDA-231 is >6-8 weeks. It is important to show that Adenoviral infected cells continue to express transgenes during this window “*

For these assays, we used cell lines stably expressing lentivirus.

7. *“The authors show that phosphorylated T367 EZH2 interacts with vinculin, however the mechanistic*

aspects are not sufficiently explored. The downstream mediators of phosphorylated T367 EZH2 need to be elucidated. For example, where does EZH2 bind vinculin? Does EZH2 impact vinculin activation, binding partners or localization? Given that pEZH2 incorporates in PRC2, why are PRC2 components missing in Fig 7b “

We appreciate the comment, and as suggested by the reviewer we have investigated these mechanistic aspects in depth in the revised paper. The new data show that pEZH2-vinculin interaction is direct and robust (KD: 15 ± 0.7 nM), occurs in the cytoplasm, induces vinculin phosphorylation at Y100, and is important for p-vinculin (Y100) localization at sites of focal adhesions; an event required for vinculin activation and enhanced migration (Auernheimer et al., 2015; Golji et al., 2012). We were unable to find that pEZH2-vinculin interaction changed vinculin stability or binding to vinculin partners talin, alpha-actinin, F-actin, or alpha-catenin (data not shown). We further found that pEZH2(T367) binding to vinculin and ensuing vinculin phosphorylation are dependent on activation of p38 MAPK. These data have been incorporated in **new Fig. 7b-e and new Suppl. Fig. 6d-f**. Please see response 3 to reviewer 1 for more details.

As noted by the reviewer, we observed that pEZH2(T367) has the capacity to bind other core PRC2 members by co-IP and PLA assays in breast cancer cells (**Suppl. Fig. 5b and 6e**), and it was previously shown that phosphorylation is not required for EZH2 to incorporate into PRC2 (Palacios et al., 2010). From our PLA data in **Fig. 7a**, we have not detected an interaction between vinculin with EED or SUZ12. Together, these data allow us to conclude that EZH2 interacts with vinculin independent of PRC2, and that pEZH2-vinculin interaction leads to vinculin Y100 phosphorylation. Supporting this observation, we performed an in vitro methyltransferase assay and were unable to detect additional sites of methylation on vinculin after it was incubated with recombinant PRC2 (data not shown). Future studies in our lab will investigate in detail how pEZH2(T367)-vinculin interaction promotes the effects of pEZH2(T367) in breast cancer migration, invasion, and metastasis which we believe is outside the scope of the present study.

8. *“It is important to determine the relationship between pEZH2 and H3K27me3 and how does this impact carcinogenesis. Also in this context, it’s important to discuss the authors previous findings that EZH2 inhibition impacts p38 signaling.”*

In our experiments using pharmacological or genetic inhibition of p38, we have found an inverse correlation between pEZH2 and H3K27 trimethylation, which is supported by our data on invasive carcinoma samples showing a significant inverse association between pEZH2 and H3K27me3 levels (**Fig. 2d-e**). The data are also supported by our new fractionation studies (**new Suppl. Fig. 3b**), which show a near complete absence of pEZH2 in the chromatin-bound fraction of breast cancer cells, and significant enrichment in the cytoplasmic and soluble nuclear fractions compared to total EZH2.

To investigate the role pEZH2 in the cytoplasm, we performed knockdown-rescue experiments with WT-EZH2 and Δ NLS-EZH2 and found that only WT-EZH2, and not Δ NLS-EZH2 was able to rescue H3K27me3 (**new Suppl. Fig. 3c quantitation**). Despite this inability to rescue H3K27me3, Δ NLS was able to bind to SUZ12 and EED and was sufficient to rescue the migratory and invasive phenotype of breast cancer cells (**Fig. 3c-f and new Suppl. Fig. 3d**), suggesting that cytoplasmic EZH2 may exert H3K27me3-independent functions. Together, these data support the hypothesis that EZH2 phosphorylation at T367 results in attenuated PRC2-dependent methylation of H3K27.

9. *“Ref #6 does not show that EZH2 and low H3K27me3 correlate with poor prognosis of ER- breast cancers, as the authors state.*

We appreciate the reviewer’s comments and apologize for the confusion. We agree that in Ref#6 (Bae et al., 2015), the authors find that the combination of high EZH2 and low H3K27me3 portends the worst survival across all 146 cases of invasive carcinoma in their study, independent of ER status. In

Ref#7, the authors find an inverse correlation between EZH2 and H3K27me3, which appears to be particularly significant for triple-negative in basal-like breast cancers (Holm et al., 2012). The revised manuscript references these studies appropriately.

Ref # 16 Placios et al does not show that p38 contributes to T367 phosphorylation as the authors state. It shows that T372 phosphorylation is impacted.”

The threonine in this study (T367) and that in Palacios et al., (T372) refer to the same threonine and result from differences in numbering depending on which isoform is used as a reference. In humans, the canonical EZH2 isoform is 746 amino acids, although there is also a 751 amino acid isoform (Isoform 2 in UniProtKB/Swiss-Prot) that has been used to position loss of function mutations in B-cell lymphomas and germline mutations leading to Weaver syndrome (Gall Trošelj et al., 2016). The five amino acid difference between the two isoforms comes from the amino acids corresponding to 297-298. The canonical sequence amino acids, HP, are HRKCNYS in isoform 2. This five amino acid difference results in different numbering of the same threonine studied in this paper (367) and the threonine in Palacios et al (372), depending on which isoform is used as a reference. Indeed, the T367 numbering has been used previously in reference to this same site by (Ko et al., 2016). The pBabe myc-EZH2 and FLAG-EZH2 constructs used in this paper correspond to the 746 aa canonical sequence, thus we used the T367 number. Of note, the human pBabe constructs used in Palacios et al. (2010) are referenced in a previous publication (Caretta et al., 2004) and appear to originate from (Varambally et al., 2002), which corresponds to the 746 aa canonical isoform. We performed Sanger sequencing to validate the 746 aa identity of the constructs used in this study.

10. *“Please quantify western blot data in Fig 1C.”*

As suggested, we have quantified the western blot data.

11. *“For functional analysis, SUM159 will be used as it expressed increased pEZH2. However, data in Fig 1c is from SUM149.”*

We thank the reviewer for pointing this error. We have corrected it in the revised manuscript, and have also added multiple experiments using SUM159 and MDA-MB-468 cells.

12. *“There is some discussion in the literature regarding specificity issues with p38 inhibitors including the ones used in this study.”*

We agree with the comment, and have complemented our experiments using shRNA to knockdown p38 in addition to the inhibitor.

13. *“In Fig 4A, Ad EZH2 and T367A show higher molecular weight compared to Wt EZH2. Myc Tag alone is unlikely to make such a difference around 100kD.”*

We observe this difference in size with all the cell lines we used. We have checked our vector (pBabe) which contains the 3myc-tag EZH2 pcDNA insert from (Varambally et al., 2002). It is most likely that the resolution of the bands is due to the percentage of gel that we use (gradient 4-12%), which resolves that area well, allowing to see the size difference of the 3myc-tag. We have observed this difference with both lentivirus and adenovirus, and in previous publications with this insert (Moore et al., 2013).

References cited in this letter.

- Auernheimer, V., Lautscham, L.A., Leidenberger, M., Friedrich, O., Kappes, B., Fabry, B., and Goldmann, W.H. (2015). Vinculin phosphorylation at residues Y100 and Y1065 is required for cellular force transmission. *J Cell Sci* *128*, 3435-3443.
- Bae, W.K., Yoo, K.H., Lee, J.S., Kim, Y., Chung, I.-J., Park, M.H., Yoon, J.H., Furth, P.A., and Hennighausen, L. (2015). The methyltransferase EZH2 is not required for mammary cancer development, although high EZH2 and low H3K27me3 correlate with poor prognosis of ER-positive breast cancers. *Molecular Carcinogenesis* *54*, 1172-1180.
- Caretti, G., Di Padova, M., Micales, B., Lyons, G.E., and Sartorelli, V. (2004). The Polycomb Ezh2 methyltransferase regulates muscle gene expression and skeletal muscle differentiation. *Genes & Development* *18*, 2627-2638.
- Chen, Z., and Cole, P.A. (2015). Synthetic approaches to protein phosphorylation. *Curr Opin Chem Biol* *28*, 115-122.
- Gall Trošelj, K., Novak Kujundzic, R., and Ugarkovic, D. (2016). Polycomb repressive complex's evolutionary conserved function: the role of EZH2 status and cellular background. *Clinical Epigenetics* *8*, 55.
- Golji, J., Wendorff, T., and Mofrad, M.R. (2012). Phosphorylation primes vinculin for activation. *Biophys J* *102*, 2022-2030.
- Holm, K., Grabau, D., Lövgren, K., Aradottir, S., Gruvberger-Saal, S., Howlin, J., Saal, L.H., Ethier, S.P., Bendahl, P.-O., Stål, O., *et al.* (2012). Global H3K27 trimethylation and EZH2 abundance in breast tumor subtypes. *Molecular Oncology* *6*, 494-506.
- Ko, H.-W., Lee, H.-H., Huo, L., Xia, W., Yang, C.-C., Hsu, J.L., Li, L.-Y., Lai, C.-C., Chan, L.-C., Cheng, C.-C., *et al.* (2016). GSK3 β ; inactivation promotes the oncogenic functions of EZH2 and enhances methylation of H3K27 in human breast cancers. *Oncotarget* *7*.
- Moore, H.M., Gonzalez, M.E., Toy, K.A., Cimino-Mathews, A., Argani, P., and Kleer, C.G. (2013). EZH2 inhibition decreases p38 signaling and suppresses breast cancer motility and metastasis. *Breast Cancer Research and Treatment* *138*, 741-752.
- Pal, A., Huang, W., Li, X., Toy, K.A., Nikolovska-Coleska, Z., and Kleer, C.G. (2012). CCN6 modulates BMP signaling via the Smad-independent TAK1/p38 pathway, acting to suppress metastasis of breast cancer. *Cancer Res* *72*, 4818-4828.
- Palacios, D., Mozzetta, C., Consalvi, S., Caretti, G., Saccone, V., Proserpio, V., Marquez, V.E., Valente, S., Mai, A., Forcales, S.V., *et al.* (2010). TNF/p38 α /polycomb signaling to Pax7 locus in satellite cells links inflammation to the epigenetic control of muscle regeneration. *Cell Stem Cell* *7*, 455-469.
- Skinner, J.J., Wang, S., Lee, J., Ong, C., Sommese, R., Sivaramakrishnan, S., Koelmel, W., Hirschbeck, M., Schindelin, H., Kisker, C., *et al.* (2017). Conserved salt-bridge competition triggered by phosphorylation regulates the protein interactome. *Proc Natl Acad Sci U S A* *114*, 13453-13458.

Su, I.h., Dobenecker, M.-W., Dickinson, E., Oser, M., Basavaraj, A., Marqueron, R., Viale, A., Reinberg, D., Wülfing, C., and Tarakhovsky, A. (2005). Polycomb Group Protein Ezh2 Controls Actin Polymerization and Cell Signaling. *Cell* *121*, 425-436.

Tosatto, A., Sommaggio, R., Kummerow, C., Bentham, R.B., Blacker, T.S., Berecz, T., Duchon, M.R., Rosato, A., Bogeski, I., Szabadkai, G., *et al.* (2016). The mitochondrial calcium uniporter regulates breast cancer progression via HIF-1alpha. *EMBO Mol Med* *8*, 569-585.

Varambally, S., Dhanasekaran, S.M., Zhou, M., Barrette, T.R., Kumar-Sinha, C., Sanda, M.G., Ghosh, D., Pienta, K.J., Sewalt, R.G.A.B., Otte, A.P., *et al.* (2002). The polycomb group protein EZH2 is involved in progression of prostate cancer. *Nature* *419*, 624-629.

Zhang, K., Corsa, C.A., Ponik, S.M., Prior, J.L., Piwnica-Worms, D., Eliceiri, K.W., Keely, P.J., and Longmore, G.D. (2013). The collagen receptor discoidin domain receptor 2 stabilizes SNAIL1 to facilitate breast cancer metastasis. *Nat Cell Biol* *15*, 677-687.

REVIEWERS' COMMENTS:

Reviewer #1 (Remarks to the Author):

In this revision, the authors addressed most of my concerns and therefore I believe it is now suitable for publication in Nature Communications. However I do think the authors need to discuss the potential caveats/alternative explanations of the biological function of the EZH2/Vinculin interaction. As it was merely showed by an IP experiment without further characterization such as sizing fractionation, interactive domain mapping..etc.